# Resolving chaperone-assisted protein folding on the ribosome at the peptide level

Thomas E. Wales [1,6], Aleksandra Pajak[2,6], Alžběta Roeselová[2], Santosh Shivakumaraswamy[2], Steven Howell[3], Svend Kjær [4], F. Ulrich Hartl [5], John R. Engen [1]✉ & David Balchin [2]✉

Protein folding in vivo begins during synthesis on the ribosome and is modulated by molecular chaperones that engage the nascent polypeptide. How these features of protein biogenesis influence the maturation pathway of nascent proteins is incompletely understood. Here, we use hydrogen–deuterium exchange mass spectrometry to define, at peptide resolution, the cotranslational chaperone-assisted folding pathway of *Escherichia coli* dihydrofolate reductase. The nascent polypeptide folds along an unanticipated pathway through structured intermediates not populated during refolding from denaturant. Association with the ribosome allows these intermediates to form, as otherwise destabilizing carboxy-terminal sequences remain confined in the ribosome exit tunnel. Trigger factor binds partially folded states without disrupting their structure, and the nascent chain is poised to complete folding immediately upon emergence of the C terminus from the exit tunnel. By mapping interactions between the nascent chain and ribosomal proteins, we trace the path of the emerging polypeptide during synthesis. Our work reveals new mechanisms by which cellular factors shape the conformational search for the native state.

Our understanding of the physical principles underlying the refolding of small proteins in isolation is continually improving[1]. De novo protein folding in vivo differs in that it begins in the context of translation. Unlike folding from denaturant, folding on the ribosome is coupled to vectorial synthesis (proceeding from the amino terminus to the C terminus), such that not all elements of the nascent protein chain (NC) are simultaneously available for folding. As a result, structural motifs or domains may form sequentially during translation[2–4], a mechanism that can mitigate interdomain misfolding[5–7]. The ribosome also places physical constraints on folding. The exit tunnel, which is narrow and negatively charged, limits the conformational space initially accessible to the nascent polypeptide[8–10], and detailed investigations of several model proteins have shown that proximity to the ribosome surface can thermodynamically destabilize complete domains, thereby delaying the onset of folding[11–15]. Nevertheless, there is abundant indirect evidence for partial folding of incomplete domains at intermediate points during translation[3,9,16–20]. How cotranslational folding proceeds during domain synthesis is incompletely understood.

The ribosome associates with additional factors that regulate folding. Most prominent among these in bacteria is the highly abundant ribosome-bound chaperone Trigger factor (TF), which engages the majority of nascent *E. coli* proteins[21]. Although not essential under normal growth conditions, TF deletion is lethal in the absence of buffering by the Hsp70 chaperone DnaK[22]. Various, sometimes paradoxical functions have been ascribed to TF. These include inhibiting aggregation[23], promoting folding and assembly[24,25], delaying cotranslational assembly[26], destabilizing folded domains[27] and favoring post-translational folding[28,29]. How exactly chaperones modify de novo folding on the ribosome is unclear.

[1]Department of Chemistry & Chemical Biology, Northeastern University, Boston, MA, USA. [2]Protein Biogenesis Laboratory, The Francis Crick Institute, London, UK. [3]Proteomics Science Technology Platform, The Francis Crick Institute, London, UK. [4]Structural Biology Science Technology Platform, The Francis Crick Institute, London, UK. [5]Department of Cellular Biochemistry, Max Planck Institute of Biochemistry, Martinsried, Germany. [6]These authors contributed equally: Thomas E. Wales, Aleksandra Pajak. ✉e-mail: j.engen@northeastern.edu; david.balchin@crick.ac.uk

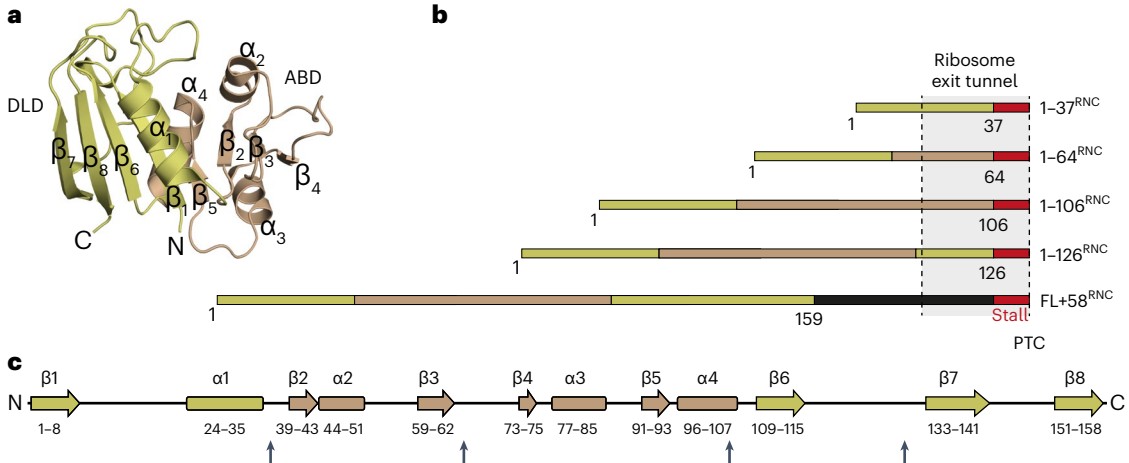

**Fig. 1 | Design of DHFR RNCs. a**, Structure of *E. coli* DHFR (PDB 5CCC). **b**, Schematic illustrations of stall-inducing constructs. The stall site is indicated in red, and subdomains are colored gold (discontinuous loop subdomain, DLD) or bronze (adenosine binding subdomain, ABD), with the artificial linker in FL + 58[RNC] in black. The region of the NC expected to span the exit tunnel is indicated. PTC, peptidyl transfer center. **c**, Secondary structure elements in DHFR, colored according to subdomains as in **b**. Sites where the stall sequence was inserted are marked with arrows.

The inherent structural heterogeneity of the nascent protein, especially in the context of the size and complexity of the ribosome (with associated chaperones), poses a substantial technical challenge to probing local NC conformation. As a result, folding pathways on the ribosome are ill-defined compared to refolding from denaturant, and whether translation fundamentally modifies the mechanism of protein maturation remains controversial[16,30–34]. Here, we use proteomics, biophysical measurements and hydrogen–deuterium exchange (HDX) mass spectrometry (MS) to study cotranslational protein folding at the peptide level. Our approach resolves subtle local conformational differences between ribosome-bound biogenesis intermediates, while simultaneously reporting on the dynamic behavior and interactions of ribosomal proteins and bound chaperones.

Using dihydrofolate reductase (DHFR) as a model, we show that the ribosome grants the NC access to an efficient folding route that is inaccessible during refolding from denaturant. Although the vectorial character of protein synthesis prevents simultaneous folding of all elements of the DHFR β-sheet, a central subdomain in DHFR behaves as a novel independent folding unit during translation. This is enabled by association with the ribosome and occurs while the NC is bound by the chaperone TF. Together, our data show how the ribosome and TF collaborate to define the structural progression of a nascent protein.

## Results

### Preparation of cotranslational folding intermediates

As a model for studying protein biogenesis, we used *E. coli* DHFR, an essential oxidoreductase enzyme with close homologs in all domains of life[35]. DHFR is a single-domain monomeric protein comprising 159 amino acids (aa), with a central discontinuous eight-stranded β-sheet and four flanking α-helices (Fig. 1a,c). During refolding from denaturant in vitro, DHFR undergoes global collapse on the microsecond timescale, and the order of strands in the β-sheet motif is established within 6 ms[36]. Importantly, folding of DHFR is substantially faster than its synthesis on the ribosome, which would require ~8–16 s (~10–20 codons translated per second[37]). Stalling the ribosome is therefore not expected to distort NC conformational sampling that occurs during normal translation.

To sample DHFR cotranslational folding intermediates along the pathway of vectorial synthesis, we prepared stalled ribosome–NC complexes (RNCs) representing snapshots of folding in vivo (Fig. 1b)[38–40]. The NC sequence was truncated at the boundaries between discrete structural motifs, also considering the annotation of DHFR 'subdomains' (adenosine binding subdomain, ABD (residues 38–106);

discontinuous loop subdomain, DLD (residues 1–37 and 107–159)). Translation was stalled by encoding, C-terminal to each DHFR fragment, an 8 aa ribosome stall-inducing sequence that is resistant to folding-induced release[41]. As a control, we prepared a construct consisting of the complete DHFR sequence separated from the stall site by an unstructured C-terminal linker of 50 aa, such that the full-length DHFR is a total of 58 residues from the peptidyl transfer center (denoted FL + 58[RNC]). Prior force profile analyses suggested that DHFR stalled at this linker length can fold into a conformation capable of binding the inhibitor methotrexate[17]. Note that ~30 residues in an extended conformation are required to span the exit tunnel[42].

RNCs purified from *E. coli* were homogeneous, stable and insensitive to puromycin, indicating that the ribosomes were accurately stalled (Extended Data Fig. 1a–f). MS confirmed the presence of the NC and all ribosomal proteins at close to the expected stoichiometries, as well as the absence of significant contamination (Extended Data Fig. 2a,b). The exception was the chaperone TF, which copurified predominantly with RNCs containing the first 106 or 126 residues of DHFR (1–106[RNC] and 1–126[RNC]; Extended Data Figs. 1c and 2c).

Purified RNCs were exposed to deuterated buffer for 10 s, 100 s or 1,000 s and the entire system was digested into peptides. The peptide mixture was separated by liquid chromatography and ion mobility, and the relative deuterium uptake of peptides from all proteins in the system was measured by mass spectrometry (Fig. 2a and Supplementary Data 1). The degree of deuterium incorporation indicates protein conformation, as backbone hydrogens are protected from exchange when involved in stable secondary structure, buried in the core of a folded protein or at a protein–protein interface[43]. Comparative HDX measurements are therefore a sensitive probe of the local environment of amide hydrogens. Although different factors can influence HDX rates, quantitative comparison with appropriate reference samples can readily distinguish folded versus unfolded and native versus non-native states at the peptide level[44–47].

We first analyzed FL + 58[RNC], in which DHFR was expected to be completely emerged from the ribosome and folded. A total of 34 peptides from the NC covering 94% of the sequence could be followed, despite the analytical complexity of the system (Extended Data Fig. 3a). For comparison, we analyzed isolated DHFR (here called FL DHFR). Deuterium uptake for DHFR peptides was almost identical between FL + 58[RNC] and FL DHFR (Fig. 2b, Extended Data Fig. 3b and Supplementary Data 1), indicating that DHFR, fully translated yet still coupled to the ribosome, is essentially natively folded. Two exceptions were subtle

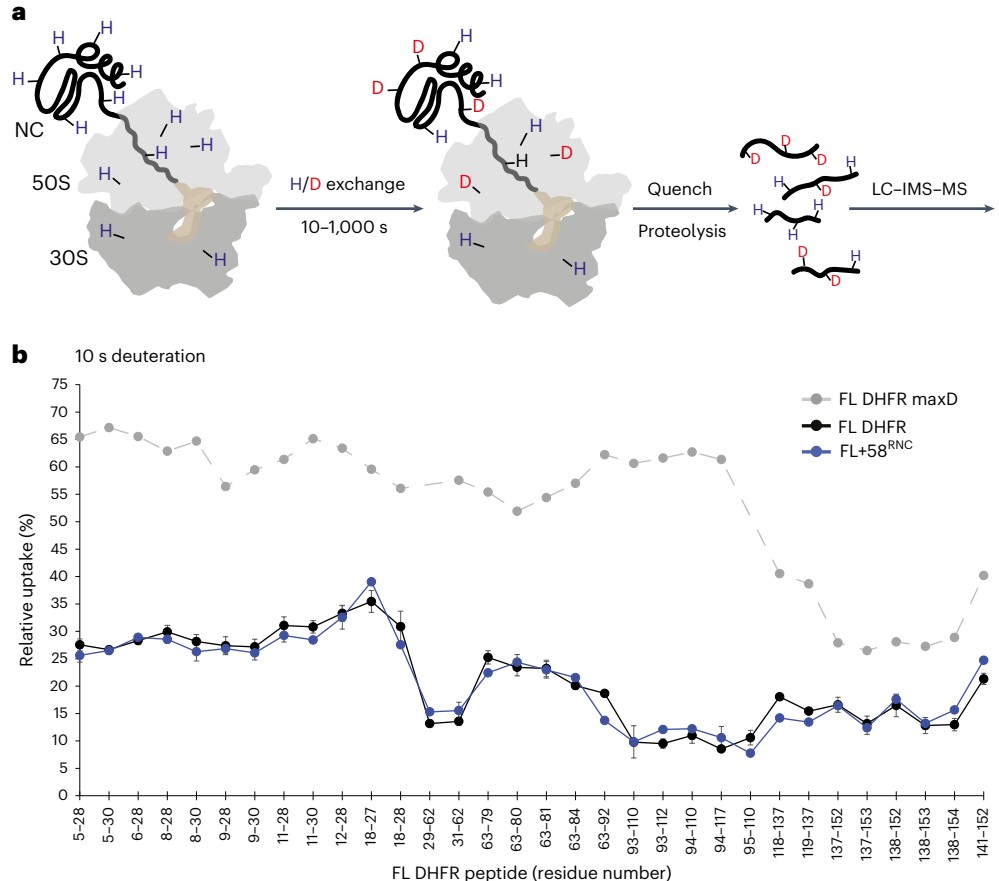

**Fig. 2 | HDX MS of RNCs. a**, Schematic of HDX MS experiment. Purified RNCs are deuterated for 10 s, 100 s or 1,000 s before the reaction is quenched and the entire system is digested into peptides. Peptides are resolved in solution (LC, liquid chromatography) and in the gas phase (IMS, ion-mobility spectrometry), followed by analysis of deuterium incorporation by MS. **b**, Relative deuterium uptake of DHFR peptides after 10 s exposure to deuterium as a percentage of the maximum possible exchange, for isolated DHFR (FL DHFR) and $FL + 58^{RNC}$. A maximally deuterated control sample (FL DHFR maxD) is shown as a reference. Lines are a guide for the eye only. Data are presented as mean values of two to four replicates. Error bars, s.d.; $n = 3$ or $n = 4$ independent experiments. See also Extended Data Fig. 3 and Supplementary Data 1.

protection from exchange of peptide 63–92 in the NC, and deprotection of peptide 94–117, both apparent at longer deuterium exposure times. These differences may arise from weak interactions between the ribosome surface and folded DHFR, discussed below. Overall, these data demonstrate that our approach allows for the analysis of HDX in an extremely complex mixture and can accurately report on local NC conformation even in the background of the entire ribosomal protein complement.

**Cotranslational folding pathway of DHFR**

To define the folding dynamics of DHFR on the ribosome, we compared peptide-resolved HDX at different NC lengths. FL DHFR and $FL + 58^{RNC}$ served as fully folded references, and deuterium uptake at three exchange times provided a fingerprint for the native conformation. We initially focused on the N-terminal region including part of strand β1 in the DLD (residues 5–30), which presented a set of overlapping peptides common to all RNCs. As a representative example, deuteration of peptide 9–28 as a function of NC length and deuterium exposure time is shown in Fig. 3a. Note that the same behavior was observed in multiple peptides covering this region of DHFR (Fig. 3b). At short chain lengths ($1–37^{RNC}$), the N-terminal region is protected relative to the maximally deuterated control but is readily distinguishable from the same sequence in native DHFR, which was much more protected at 10 s deuteration. Non-native protection at short chain lengths may be a result of interactions with the ribosome or reflect transient non-native structure stabilized by the exit tunnel[4]. Although

substantially deprotected relative to native DHFR, the N-terminal region was not completely deuterated in the RNCs, as judged by comparison to deuteration of a synthetic peptide comprising residues 1–37 ($1–37^{peptide}$), which was maximally exchanged at the earliest labeling time. Elongation of the NC to extend the N terminus fully beyond the exit tunnel ($1–64^{RNC}$) resulted in further deuteration. The N-terminal region remained highly deuterated with little variation in uptake, even as the NC extended to 106 and 126 residues during synthesis of the ABD. Folding of the DLD is therefore delayed until the complete subdomain is available and the C terminus is released from the ribosome, or artificially extended beyond the tunnel through a linker as in $FL + 58^{RNC}$. Notably, this folding pathway is different than the in vitro pathway of refolding of DHFR from denaturant, in which the central eight-stranded β-sheet that spans the two subdomains is established early in the folding pathway[48].

Extending this analysis to additional peptides across the RNCs, each reporting local folding information, allowed us to reconstruct the complete cotranslational folding pathway of DHFR. Peptide-resolved HDX for each different length RNC is shown in Fig. 3b, Extended Data Fig. 4a,b and Supplementary Data 1, and folding information is mapped onto the structure of native DHFR in Fig. 3c. Although not all peptides were uniformly detected across stalled RNCs of different lengths, clear patterns of HDX protection corresponding to NC folding were apparent. As described above, the N-terminal region including strand β1 is not folded when initially exposed outside the exit tunnel ($1–37^{RNC}$), with deuteration levels close to the maximally

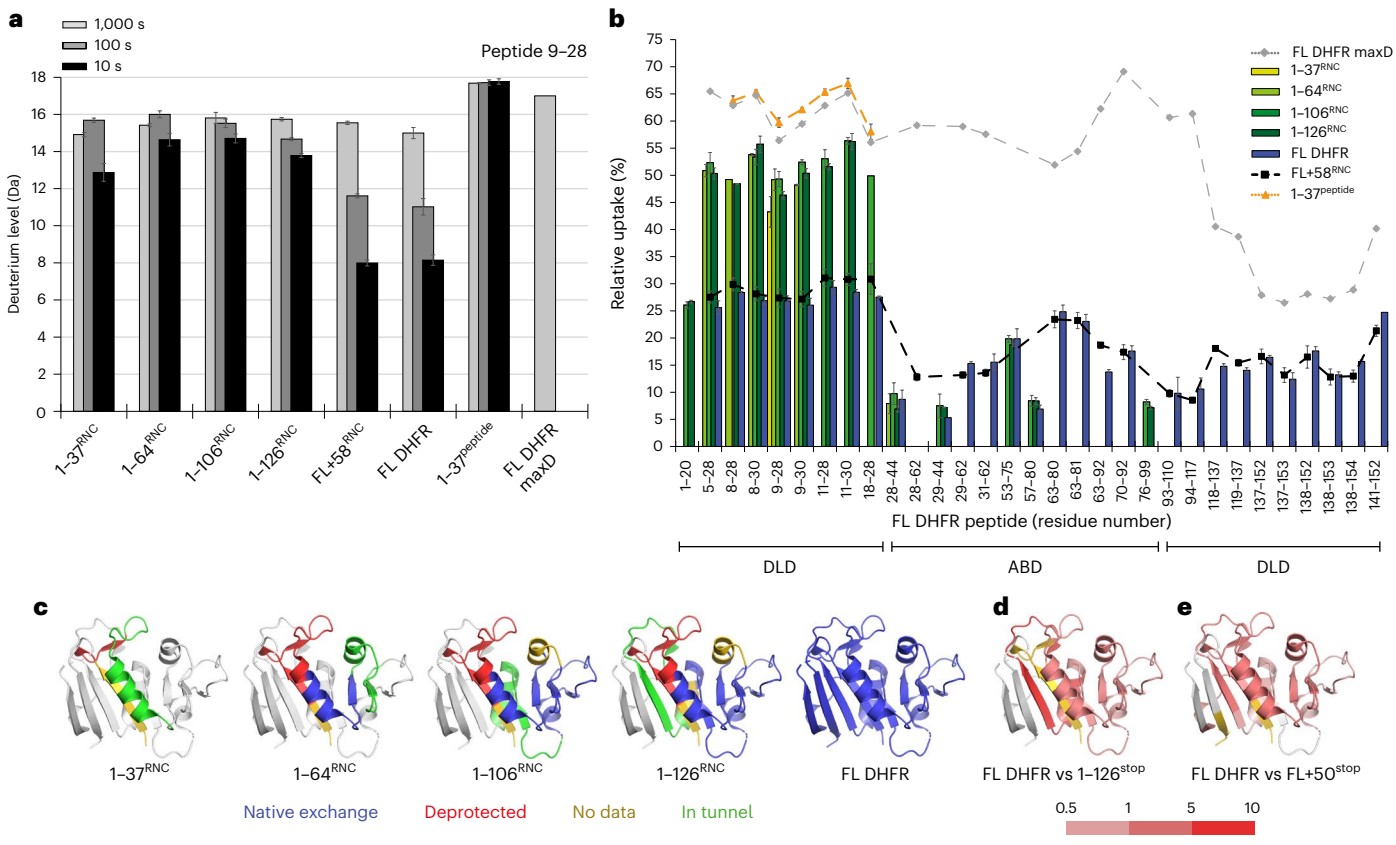

**Fig. 3 | Conformational dynamics of nascent DHFR on the ribosome.**
**a**, Deuterium incorporation into peptide 9–28 of DHFR as a function of deuterium exposure time and NC length. Data are presented as mean values of two to four replicates. Error bars, s.d.; $n = 3$ or $n = 4$ independent experiments. **b**, Relative deuterium uptake of DHFR peptides after 10 s exposure to deuterium. Peptides belonging to the DLD or ABD subdomains are indicated. Data are presented as mean values of two to four replicates. Error bars, s.d.; $n = 3$ or $n = 4$ independent experiments. **c**, Peptide-resolved folding of nascent DHFR, illustrated on the structure of native DHFR (PDB 5CCC). Peptides with the same deuterium uptake (within 0.5 Da) as the native controls (FL DHFR or FL + 58$^{RNC}$) are colored blue, while peptides that are deprotected relative to the controls are

colored red. Regions for which we did not measure HDX are colored gold. The C-terminal 22 aa in each construct, expected to be occluded in the exit tunnel at that chain length, is colored green. The part of DHFR that is not yet synthesized at each chain length is colored white. **d**, Difference in deuterium uptake between FL DHFR and 1–126$^{stop}$ after 10 s deuteration. Darker red indicates more deuterium in 1–126$^{stop}$ relative to FL DHFR (increasing deprotection). Residues 127–159, not present in 1–126$^{stop}$, are colored white. **e**, Difference in deuterium uptake between FL DHFR and FL + 50$^{stop}$ after 100 s deuteration. Darker red indicates increasing deprotection of FL + 50$^{stop}$ relative to FL DHFR. Peptides that do not differ in exchange are colored white. See also Extended Data Fig. 4 and Supplementary Data 1.

deuterated control. In 1–64$^{RNC}$, residues 25–36 emerge from the tunnel and fold into a native helix α1 belonging to the DLD, while the N-terminal region remains unstructured. Synthesis of the remainder of the ABD (1–106$^{RNC}$) allows β2–β4 to exit the ribosome and acquire native-like protection from HDX. This protection was not a result of association with TF, as discussed in subsequent sections. Folding of the ABD completes when 126 residues of DHFR have been synthesized (1–126$^{RNC}$), allowing β5 and α4 to coalesce with the remainder of the ABD outside the exit tunnel. At this point, β6 is still occluded in the tunnel, precluding folding of the adjacent β1 in the DLD. Synthesis and release of the C-terminal strand β8 from the ribosome triggers the final folding of the DLD, including the N-terminal segments (FL DHFR and FL + 58$^{RNC}$).

**The ribosome stabilizes a cotranslational folding intermediate**

Unlike the DLD, peptides corresponding to the β-sheet of the ABD became progressively protected from HDX during translation, requiring neither the sequence context of FL DHFR nor even the complete subdomain (Fig. 3c). By contrast, previous work has shown that fragments of DHFR produced by chemical cleavage are disordered in isolation[49]. To test whether the ribosome modulates the conformation of the

nascent ABD, we expressed isolated fragments corresponding to the ABD in *E. coli*. Only the longest fragment, consisting of residues 1–126 (herein 1–126$^{stop}$), was soluble, and this was contingent on maintaining it as a fusion protein, with monomeric ultrastable GFP (muGFP[50]) at the N terminus. HDX MS showed that 1–126$^{stop}$ was substantially deprotected relative to native DHFR (Fig. 3d and Supplementary Data 1). High levels of exchange were observed for the N-terminal region, as expected in the absence of C-terminal strands β7 and β8 that complete the DLD. The β-strands (β2–β4) and peripheral helices comprising the core of the ABD were also strongly deprotected relative to FL DHFR. The ABD, therefore, fails to fold into a stable structure in isolation, although native-like folding of this subdomain is supported on the ribosome in the context of an RNC.

We asked why incomplete chains of DHFR can fold on the ribosome but not in free solution. We noticed that residues 93–118 near the C terminus of 1–126$^{stop}$ were maximally deuterated even at short deuteration times, indicative of structural disorder (Fig. 3d and Supplementary Data 1). In the corresponding RNC, however, this sequence is sterically confined in the ribosome exit tunnel (Figs. 1b and 3c). Therefore, we considered whether, in the absence of the ribosome, a region of C-terminal disorder may destabilize folded DHFR. Previous work has shown that unstructured termini can generate an entropic

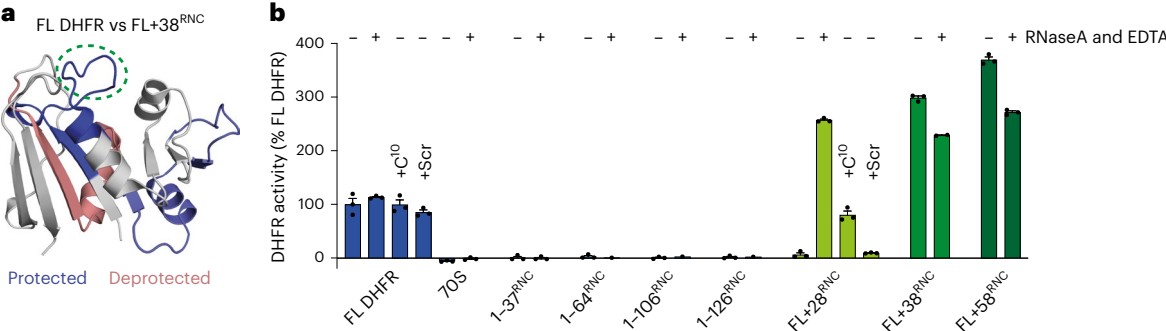

**Fig. 4 | Ribosome contacts modulate the structure of nascent DHFR without inducing unfolding. a**, Relative deuterium uptake of FL + 38$^{RNC}$ compared to FL DHFR. DHFR peptides that are protected in the RNC relative to FL DHFR, at any deuteration time point, are colored blue; deprotected peptides are colored red. The Met20 loop is indicated with a green dashed ellipse. See also Extended Data Fig. 6a and Supplementary Data 1. **b**, Oxidoreductase activity of 25 nM

FL DHFR, empty 70S ribosomes without an NC or RNCs. Where indicated, reactions are supplemented with 50 µg ml⁻¹ RNaseA and 50 mM EDTA, a peptide corresponding to the C terminus of DHFR (C$^{10}$, SYCFEILERR) or a scrambled-sequence control peptide (Scr, RFIERCELYS). Data are presented as mean values; error bars, s.d.; $n$ = 3 independent experiments.

force that modulates protein conformation[51]. To explore this idea, we used FL + 58$^{RNC}$ as a model. FL + 58$^{RNC}$ consists of FL DHFR tethered to the peptidyl transfer center (PTC) by a flexible 50 aa linker that is largely occluded in the exit tunnel (Fig. 1b). Our HDX analysis showed that DHFR is natively folded in this context and almost indistinguishable from the isolated domain (Fig. 2b and Extended Data Fig. 3b,c). We reasoned that releasing the protein from the ribosome would remove the protective effect of the exit tunnel and allow an unstructured terminus to disrupt the conformation of DHFR. To test this idea, we replaced the stalling sequence with a stop codon, resulting in FL + 50$^{stop}$. HDX MS analysis of FL + 50$^{stop}$ confirmed that the linker was highly dynamic, as expected (Extended Data Fig. 4c–e and Supplementary Data 1). Furthermore, we observed extensive deprotection in the DLD and ABD relative to native FL DHFR, consistent with increased structural dynamics (Fig. 3e and Extended Data Fig. 4c–e). Combined, our observations suggest a model whereby domain instability induced by unstructured C termini is rescued by confinement in the ribosome exit tunnel, potentially facilitating the folding of structured intermediates during translation.

### DHFR can fold close to the ribosome

Proximity to the ribosome surface thermodynamically destabilizes some full-length nascent proteins relative to the native state of the ribosome[11–13,15], prompting us to consider whether tethering to the ribosome might alter the conformation of FL DHFR. HDX analysis of FL + 58$^{RNC}$ did not reveal substantial differences in conformation compared to isolated DHFR (Fig. 2b). We therefore prepared an RNC with a 38-residue linker (FL + 38$^{RNC}$), designed to bring DHFR closer to the ribosome surface. The folded core of DHFR in FL + 38$^{RNC}$ was similar in protection to isolated DHFR, although peripheral loops and part of the DLD were affected (Fig. 4a, Extended Data Fig. 5 and Supplementary Data 1). Furthermore, both FL + 58$^{RNC}$ and FL + 38$^{RNC}$ were enzymatically active (Fig. 4b). Therefore, close proximity to the ribosome exit tunnel does not substantially disrupt the conformation of folded wild-type DHFR.

Remarkably, FL + 38$^{RNC}$ and FL + 58$^{RNC}$ were at least twofold to fourfold more active than isolated DHFR (Fig. 4b and Extended Data Fig. 6a,b). $V_{max}$ and, importantly, $K_M^{DHF}$ were higher for the RNC, indicating that this effect is not explained by errors in measurement of enzyme concentration (Fig. 4b and Extended Data Fig. 6d). The results were also not explained by tube adsorption (Extended Data Fig. 6a). The extent of activity stimulation at fixed substrate concentration may have been underestimated because RNC concentration was calculated by assuming 100% occupancy of NCs on ribosomes (See Methods). This was bona fide DHFR activity, as it was fully inhibited by methotrexate, and

neither empty ribosomes nor intermediate-length RNCs were active (Fig. 4b and Extended Data Fig. 6e). We observed the same effect for an RNC with a different linker sequence, indicating that hyperactivity is not an artefact of linker chemistry (Extended Data Fig. 6c). Notably, the region near the N terminus of DHFR that is protected in FL + 38$^{RNC}$ includes the 'Met20 loop' at the folate binding site of DHFR, the dynamics of which are known to strongly influence catalysis[35] (Fig. 4a). Thus, physical interactions with the ribosome may alter the active site of DHFR without inducing global unfolding.

To test the threshold linker length for folding of DHFR on the ribosome, we prepared FL + 28$^{RNC}$. This RNC was devoid of oxidoreductase activity (Fig. 4b) and copurified with TF (Extended Data Fig. 2c). TF was not responsible for the lack of activity, as FL + 28$^{RNC}$ purified from a TF-free background was also inactive (Extended Data Fig. 6c). We reasoned that a 28 aa linker is insufficient to expose the C terminus of DHFR outside the exit tunnel, precluding folding of the DLD. Indeed, DHFR activity could be restored by releasing the NC from FL + 28$^{RNC}$ with EDTA and RNase (Fig. 4b). This observation is consistent with our interpretation, based on HDX MS, that DHFR folding completes post-translationally when the C terminus emerges from the tunnel (Fig. 3a–c). To test whether DHFR in FL + 28$^{RNC}$ is folding-competent before release from the ribosome, we added a peptide comprising the C-terminal 10 aa of DHFR (peptide C$^{10}$) in *trans*. Peptide C$^{10}$ activated FL + 28$^{RNC}$ in a concentration-dependent manner, whereas a scrambled-sequence control had no effect (Fig. 4b and Extended Data Fig. 6c). None of the peptides influenced the activity of isolated FL DHFR. TF was not required for reactivation, as TF-free FL + 28$^{RNC}$ was similarly reactivated (Extended Data Fig. 6c). Peptide C$^{10}$ did not release the NC from ribosomes, nor did it displace TF (Extended Data Fig. 6f). DHFR is therefore poised to complete folding upon emergence of the C terminus from the ribosome exit tunnel, and neither close proximity to the ribosome surface nor association with TF prevent acquisition of the native state.

### Interaction of TF with DHFR on the ribosome

We next sought to determine the role of TF in the cotranslational folding of DHFR. TF is a ribosome-associated chaperone with a three-domain architecture (Fig. 5a). The ribosome binding domain (RBD) contains a conserved ribosome-interaction motif[52], and the substrate binding domain (SBD) is required for chaperone function off the ribosome[53]. The role of the peptidyl-prolyl isomerase domain (PPD) in de novo folding is enigmatic[54]. Given that chaperone–RNC complexes are highly dynamic, details of the interaction between TF and NCs have eluded structural characterization[55,56], although site-specific photocrosslinking showed that NCs contact all three domains of TF[56,57].

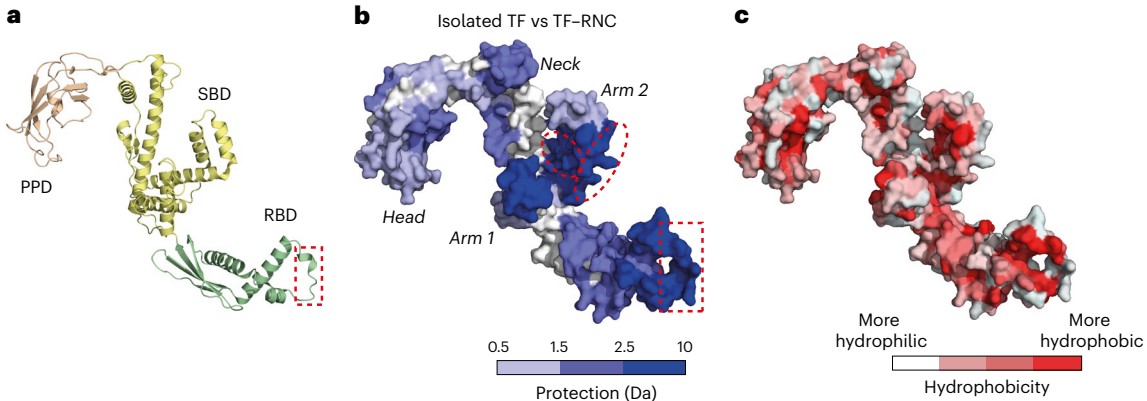

**Fig. 5 | NC interactions with TF. a,** Structure of *E. coli* TF (PDB 1W26). RBD, ribosome binding domain; SBD, substrate binding domain; PPD, peptidyl-prolyl isomerase domain. The ribosome binding motif in the RBD is indicated with a red dashed rectangle. **b,** Difference in deuterium uptake between isolated TF and TF bound to DHFR RNCs after 10 s deuteration. Darker blue indicates less deuteration of RNC-bound TF relative to isolated TF (increasing protection). The sites on arm 2 that preferentially engage 1–126$^{RNC}$ are marked by ellipses. See also Supplementary Data 1. **c,** TF surface colored according to hydrophobicity[75].

TF copurified with 1–106$^{RNC}$ and 1–126$^{RNC}$, coinciding with the synthesis of the complete ABD (Extended Data Figs. 1c and 2c). To characterize the interaction between TF and nascent DHFR, we analyzed deuterium uptake for endogenous TF in the RNC samples and compared these data to HDX MS measurements of isolated TF. We followed 181 peptides for TF in each condition, with 99.5% sequence coverage (Extended Data Fig. 7a and Supplementary Data 1). A potential confounding factor is that isolated TF is weakly dimeric (dimerization $K_d$ ~1 μM) but binds ribosomes as a monomer[57–59]. To account for this effect, we used both wild-type TF and a constitutively monomeric variant[60] as reference controls for isolated TF. HDX MS confirmed that the monomeric variant was deprotected relative to wild-type TF at sites that are normally buried by the dimer interface (Extended Data Fig. 7b and Supplementary Data 1).

Compared to isolated TF, TF that was bound to 1–106$^{RNC}$ and 1–126$^{RNC}$ was very strongly protected from HDX in the ribosome-interacting motif of the RBD, as expected[52] (Fig. 5b). We also observed protection of additional regions in all three domains of TF, which we attribute to interaction with nascent DHFR (Fig. 5b). Strong protection was observed in the two 'arms' of the SBD, with intermediate protection in the RBD, 'neck' region and catalytic site in the PPD. Analysis of the binding interface revealed a mix of hydrophobic and hydrophilic surfaces (Fig. 5c). The interaction sites in the RBD and PPD are predominantly hydrophobic. In the SBD, the interface includes the hydrophobic pocket in the crook of arm 2 as well as hydrophilic surfaces in arm 1 and the neck. The hydrophilic part of the neck situated between the arms, previously implicated in substrate engagement off the ribosome[61], was not protected from exchange. The protected sites were the same in TF bound to 1–106$^{RNC}$ and 1–126$^{RNC}$, indicating a common interaction surface. We did, however, detect increased protection of hydrophilic regions in arm 2 (residues 351–374 and 388–409) when TF was bound to the longer NC (Fig. 5B (red ellipses) and Supplementary Data 1).

### Nascent DHFR can fold while associated with TF
To determine how TF influences the folding of DHFR on the ribosome, we purified 1–106$^{RNC}$ and 1–126$^{RNC}$ from E. *coli* lacking TF (ref. 22), resulting in 1–106$^{RNCΔTF}$ and 1–126$^{RNCΔTF}$. MS confirmed that the absence of TF did not result in other chaperones (for example DnaK, DnaJ or GroEL) copurifying with the RNCs (Extended Data Fig. 2c). Pelleting assays showed that 1–126$^{RNCΔTF}$ was still competent to bind purified TF in vitro, and binding was sensitive to mutation of the ribosome-interacting motif in TF (Extended Data Fig. 8a).

To probe the conformation of the NC without TF, we analyzed the HDX behavior of nascent DHFR in 1–106$^{RNCΔTF}$ and 1–126$^{RNCΔTF}$ (Extended Data Fig. 7c and Supplementary Data 1). A set of peptides reported on the N terminus and β-strands in the ABD enabled us to compare wild-type and ΔTF RNCs. Although some marginal differences in deuterium uptake could be detected in the absence of TF, these were not consistently observed across overlapping peptides. Cotranslational folding of the ABD therefore occurs irrespective of the presence of TF, and TF binding does not explain our observation that the N-terminal region remains unfolded until release of the C terminus from the ribosome (Fig. 3c).

Considering that DHFR folding can occur in the presence of TF, we questioned what features of the NC are recognized by the chaperone. To probe the contribution of electrostatic versus hydrophobic interactions to binding, we tested the salt sensitivity of the TF–RNC interaction. TF preferred 1–106$^{RNC}$ over 1–126$^{RNC}$ when the complexes were purified under high-salt conditions (Extended Data Fig. 2c). By contrast, RNCs purified under low-salt conditions bound similar amounts of TF (Extended Data Fig. 8b). TF binding to 1–126$^{RNC}$ is therefore partially stabilized by electrostatic interactions, unlike binding to 1–106$^{RNC}$, which is predominantly mediated by hydrophobic contacts and therefore stabilized by high salt. This observation is consistent with the folding-induced burial of the ABD hydrophobic core in 1–126$^{RNC}$ (Fig. 3c) as well as the preference of this NC for binding hydrophilic surfaces on TF (Fig. 5b).

To directly test the contribution of NC folding to TF binding, we introduced destabilizing mutations[62] into 1–126$^{RNC}$ (Extended Data Fig. 8c). The mutated RNC bound more TF than wild-type 1–126$^{RNC}$ when purified under high-salt conditions, but the difference was much less pronounced when binding was reconstituted in vitro under low-salt conditions (Extended Data Fig. 8d,e). Thus, TF engages poorly folded intermediates by hydrophobic surfaces. Taken together, these observations indicate that TF uses a composite hydrophobic–hydrophilic interface to accommodate both folded and unfolded NCs, and provide indirect evidence supporting our conclusion that the ABD is natively folded in wild-type 1–126$^{RNC}$. Our low-salt conditions are in a similar range of ionic strength to the *E. coli* cytosol (~100–200 mM (ref. 63)). In vivo, TF would thus be expected to bind equally well to NCs exposing different amounts of hydrophobic surface, exploiting different types of interaction in each case.

### NC interactions with ribosomal proteins
To identify possible NC–ribosome interaction sites, we compared the HDX of ribosomal proteins in the RNCs to the same proteins in empty 70S ribosomes. We focused on a set of five ribosomal proteins (L4, L22, L23, L24 and L29) that are near or in the exit tunnel and therefore likely

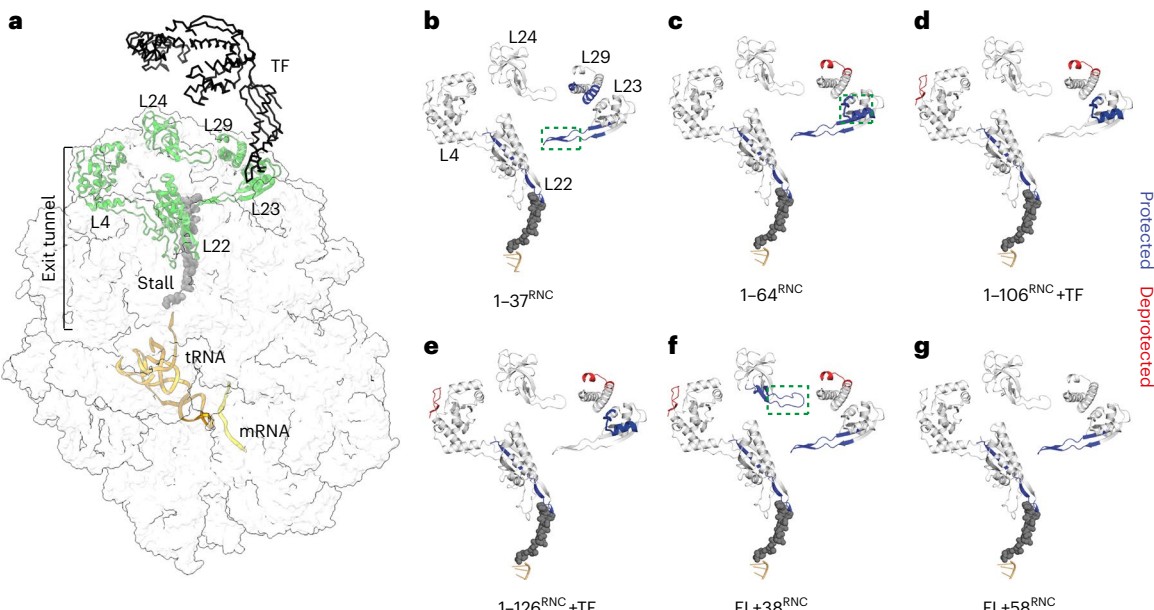

**Fig. 6 | Effect of nascent DHFR on ribosomal proteins. a**, Side view of the 70S ribosome, highlighting ribosomal proteins that line the exit tunnel and vestibule (PDB 3JBU). TF is placed onto the structure based on a previous publication[56]. The structure contains the 17-aa wild-type SecM stall-inducing sequence, although to match the length of the sequence used in this study only eight amino acids are shown. **b–g**, HDX analysis of ribosomal proteins in RNCs. Peptides that are protected from HDX in each RNC relative to empty ribosomes, at any deuteration time point, are colored blue. Deprotected peptides are colored red. The tunnel-facing loop of L23 is indicated with a green dashed rectangle in **b**, the TF docking site is indicated with a green dashed rectangle in **c** and the exposed loop of L24 is indicated with a green dashed rectangle in **f**. See also Supplementary Data 1.

to contact the emerging NC (Fig. 6a). Sequence coverage of these proteins was close to 100% (Extended Data Fig. 9a). We identified several sites of HDX protection in ribosomal proteins when NC was present, often dependent on NC length (Fig. 6b–g and Supplementary Data 1).

The lower part of the tunnel is occupied by the stall-inducing sequence. Consequently, the tunnel-exposed loop of L22, which together with L4 forms the constriction site close to the PTC, was protected from exchange at all NC lengths. The corresponding loop in L4 was not protected, consistent with structural data showing that wild-type SecM contacts L22, not L4 (refs. 64,65) (Fig. 6a). Peptide 22–42 of L29, located on the ribosome surface adjacent to the exit port, was protected only in the shortest RNC (1–37[RNC]) (Fig. 6b). Longer NCs may expose sequences that favor different interaction sites (discussed below) or preferentially interact with TF rather than L29.

Peptide 8–29 of L23, covering the binding site for TF on the ribosome surface[52], was strongly protected in 1–106[RNC] and 1–126[RNC] (Fig. 6d,e); as expected, given that these RNCs recruit TF (Extended Data Fig. 2c). However, we also detected protection of the TF docking site in 1–64[RNC], which does not engage TF (Fig. 6c). In 1–64[RNC] (but not 1–37[RNC]), the emerging NC is highly basic (pI ~9.9), perhaps facilitating electrostatic interactions with the acidic TF docking site. Direct competition between the NC and the TF-binding site on the ribosome may be an additional mechanism regulating TF recruitment to RNCs.

Peptide 61–84 of L23, forming a hairpin loop that protrudes into the exit tunnel ~50 Å from the PTC (Fig. 6b, green rectangle), was protected in all RNCs except 1–106[RNC] and 1–126[RNC], which strongly recruit TF (Extended Data Fig. 2c), suggesting the possibility of allosteric communication between the NC and TF through L23 as previously hypothesized[66]. Removing TF did not result in new protection of the L23 tunnel loop, arguing against reciprocal communication between TF and L23 (Extended Data Fig. 9b,c and Supplementary Data 1).

In FL + 58[RNC], none of the proteins surrounding the exit port were protected from HDX relative to empty ribosomes, consistent with the absence of stable interactions with the ribosome surface (Fig. 6g). However, FL + 38[RNC] protected a loop in L24 (Fig. 6f, green rectangle) that was

previously identified to interact with NCs[13], which may underlie the modulation of the DHFR conformation that we observe for this RNC (Fig. 4a).

Together, our HDX MS analysis of ribosomal proteins suggests that nascent DHFR does not generically interact with proteins comprising the exit tunnel, but rather samples a biased route during synthesis that is potentially dictated by the chemical properties of the emerging sequence. As previously described for other systems, specific NC–ribosome interactions may further modulate the pathway of cotranslational folding[10,67].

## Discussion

We analyzed the conformational dynamics of ribosome–chaperone–NC systems using HDX MS. This approach is label-free, yields local information (resolved to peptide level) and simultaneously reports on the structural dynamics of all proteins in the system.

Our results add to the range of folding scenarios previously described to occur at the ribosome[3,13,33,34,68]. We find that cotranslational folding of DHFR is neither strictly sequential (N terminus to C terminus) nor concerted (all-or-none) but rather proceeds through a combination of both mechanisms. Sequential folding of the middle subdomain poises DHFR for rapid completion of folding in a concerted post-translational step involving both the N terminus and C terminus (Fig. 7). This may be a generic mechanism exploited by proteins with discontinuous domains to minimize the delay between synthesis and acquisition of the native state.

Several studies have found evidence for differences between cotranslational and refolding intermediates[6,19,69]. During refolding in vitro, elements of the central β-sheet of DHFR fold simultaneously rather than sequentially, and intermediates with a folded ABD are not populated. The ribosome chaperones nascent DHFR in at least two ways. First, by acting as a solubility tag[70], the ribosome allows the NC to access conformations that are too aggregation-prone to persist in isolation. Second, occlusion of the dynamic C terminus in the exit tunnel prevents entropic destabilization of vulnerable folding intermediates. Disordered termini can alter the stability and dynamics of

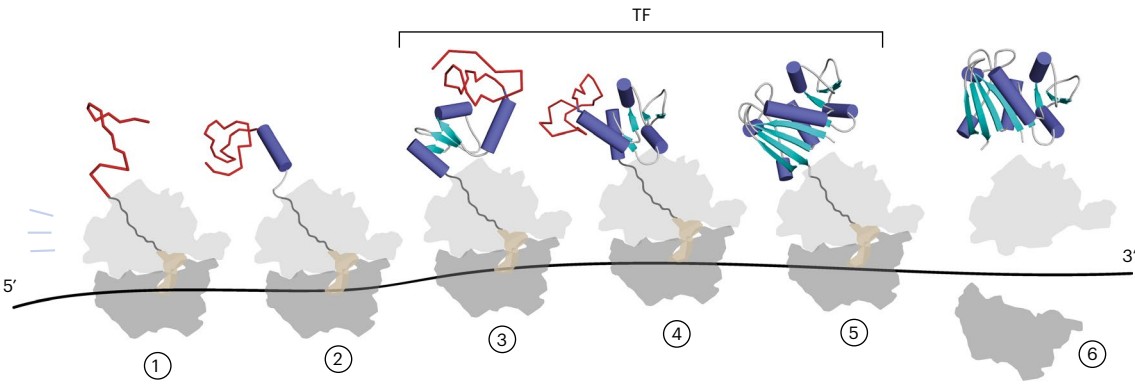

**Fig. 7 | Schematic biogenesis pathway of DHFR.** The N terminus of DHFR emerges from the ribosome in an unstructured state (red) and interacts weakly with L29 at the exit port (1). Synthesis of an additional 27 residues results in the folding of helix α1 and exposes a basic patch that contacts the surface-exposed loop of L23 (2). The remainder of the ABD folds cotranslationally and, while engaged by TF, docks at L23 (3,4). The nascent subdomain is protected from denaturation by occlusion of the succeeding sequence in the exit tunnel. Folding of the DLD, including the N-terminal strand, occurs when the C terminus is provided as a peptide in *trans* (5) or when the complete domain is released from the exit tunnel upon termination of synthesis (6).

folded proteins through excluded volume effects[51,71]. We reproduce this phenomenon for DHFR and show that the ribosome buffers entropic destabilization during protein synthesis. In this regard, the physical dimensions of the exit tunnel, which are conserved in cytosolic ribosomes[72], may have facilitated the evolution of topologically complex folds. Interactions between the NC and ribosome surface (Fig. 6b–g) are also likely to have a role in modulating the stability of cotranslational folding intermediates, as previously shown[10,73].

Although the chaperone function of TF off the ribosome is well characterized[23,24,53,61,74], its canonical role as a cotranslational chaperone is comparatively poorly understood. Here, we address two outstanding questions: considering that TF binds the majority of nascent proteins, what properties of NCs are recognized, and how does TF engagement influence the conformation of the NC? We find that NCs with little compact structure fail to engage TF, as evidenced by the lack of binding to $1\text{–}37^{RNC}$ and $1\text{–}64^{RNC}$, as well as $FL + 58^{RNC}$, which presents an unstructured linker to the chaperone. This argues against the unstructured N terminus being the driver of TF binding to intermediate-length RNCs. Instead, our data indicate that TF prefers partially ($1\text{–}106^{RNC}$, $1\text{–}126^{RNC}$) or near-fully folded (complemented $FL + 28^{RNC}$) domains (Fig. 7). Native-like folding is not, however, required for TF binding, shown by the strong recruitment of TF to an RNC exposing a destabilized subdomain. We propose that this remarkable plasticity in binding is achieved by dynamic sampling of an unusually large chaperone–NC interface that includes both hydrophilic and hydrophobic surfaces (Fig. 5). In this model, persistent binding to RNCs would require simultaneous engagement of multiple sites across the TF surface, with ribosome binding by TF contributing to overall avidity. The chemically heterogeneous binding surface accommodates both partially and fully folded domains, allowing continuous engagement of NCs as they mature. Folding may therefore occur while the NC is associated with TF, as has been suggested for refolding off the ribosome[61]. Importantly, the binding mode we describe allows TF to engage fragile folding intermediates without disrupting the incipient structure. Indeed, we observe that cotranslational folding intermediates of DHFR are essentially identical regardless of the presence of TF.

Interactions with the ribosome surface have been shown to thermodynamically destabilize several full-length nascent domains[11,12,14,15] and can stabilize folding intermediates of a full-length immunoglobulin domain relative to the native state[73]. We find that the conformational dynamics of the folded core of FL DHFR are unperturbed by proximity to the ribosome, and our peptide complementation experiments show that DHFR can fold to an enzymatically active state very close to the ribosome surface (Fig. 7). These observations are not inconsistent with previous findings. Rather, structural destabilization may be local rather than global and may not result in a substantial population of unfolded conformations at equilibrium. As sequence-specific interactions with the ribosome have been implicated in NC destabilization[14], it is also likely that any effect is protein-dependent. Our results point to the possibility that ribosome interactions may positively modulate the function of N-terminal domains in multidomain proteins, for example by promoting cofactor loading during synthesis.

Although we chose to analyze in detail only the NC and a subset of ribosomal proteins that directly engage the NC, the high-quality HDX MS dataset we describe here is comprehensive. Our approach could shed new light on other aspects of translation regulation, especially dynamic processes that have been challenging to study using conventional structural biology methods.

## Online content

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

## Methods

### Expression and purification of RNCs

Expression constructs were synthesized by Twist Biosciences. All open reading frames were cloned between a ribosome binding site (TTTGTT-TAACTTTAAGAAGGAGA) and 6×His-tag followed by a stop codon in pET21 plasmids with ampicillin resistance. The amino acid sequences encoded by each open reading frame are listed and annotated in Extended Data Table 1. Sequences are listed in Supplementary Table 1.

Plasmids were used to transform *E. coli* BL21(DE3) wild-type (NEB) or Δ*tig* (J. Christodoulou, University College London) cells. Bacteria were grown in ZYM 5052 autoinduction media[76] with 100 μg ml⁻¹ ampicillin at 37 °C for 18 h. Cells were collected by centrifugation (4,000$g$, 20 min, 4 °C) and lysed in RNC lysis buffer (Tris-HCl pH 7.5, 150 mM KCl and 10 mM MgCl$_2$) containing 1.25 mg ml⁻¹ lysozyme and subjected to 2× freeze–thaw cycles in the presence of 2.72 Kunitz units per μl DNase (QIAGEN). The soluble fraction was obtained by centrifugation at 20,000$g$ for 20 min, loaded onto a 35% sucrose cushion in high-salt RNC buffer (50 mM HEPES-KOH, pH 7.5, 12 mM Mg(OAc)$_2$, 1 M KOAc and 1 mM dithiothreitol (DTT)) and centrifuged at 4 °C for 2 h at 250,000$g$ (Beckman ultracentrifuge, TLA-110 rotor, tube no. 363305). The ribosome pellet was resuspended overnight at 4 °C in low-salt RNC buffer (50 mM HEPES-KOH, pH 7.5, 12 mM Mg(OAc)$_2$, 100 mM KOAc and 1 mM DTT). The resuspended pellet was applied to in-house-prepared GFP-clamp coupled agarose beads[77] and incubated overnight at 4 °C. RNCs were eluted by cleavage with 0.4 mg ml⁻¹ HRV 3C protease for >5 h, followed by a second round of sucrose cushion ultracentrifugation in high-salt RNC buffer (2 h, 250,000$g$). For low-salt purifications, both sucrose cushions were prepared using low-salt RNC buffer. In all cases, the final pellet was resuspended in low-salt RNC buffer and A260 was measured to estimate the RNC concentration. See also Extended Data Fig. 1a.

Where indicated, purified RNCs were incubated with either 50 μg ml⁻¹ RNaseA (NEB) and 50 mM EDTA, or 2.5 mM puromycin (Santa Cruz Biotechnology) at 20 °C for 20 min.

### Expression and purification of isolated DHFR variants

*E. coli* BL21(DE3) cells were transformed with the plasmid expressing FL DHFR from a T7 promoter (supplied as control plasmid with the NEB PURExpress kit) and selected on an LB agar plate supplemented with 100 μg ml⁻¹ ampicillin. A single colony was used to inoculate 500 ml of ZYM 5052 autoinduction media containing 100 μg ml⁻¹ ampicillin. The culture was grown for 24 h in an orbital shaker set to 37 °C. Cells were collected by centrifugation and resuspended in lysis buffer (25 mM Tris-HCl pH 7.5, 1 mM EDTA and 2 mM DTT). The resuspended cells were supplemented with 1 mM phenylmethanesulfonyl fluoride (PMSF), 0.025 units per ml benzonase, two tablets of EDTA-free protease inhibitor cocktail and then lysed by sonication. The lysate was centrifuged at 48,000$g$, 4 °C, 45 min, and the supernatant was loaded onto a 6 ml RESOURCE Q column connected to an ÄKTA Pure Protein Purification System. The bound protein was eluted with a linear gradient of 0–500 mM NaCl in lysis buffer. The eluted fractions were examined using NuPAGE 4–12% Bis-Tris gels and the fractions with pure protein were pooled, concentrated using centrifugal filters and injected into a HiLoad 16/600 Superdex 200 pg column that was equilibrated with gel filtration buffer (25 mM Tris-HCl pH 7.5, 200 mM NaCl, 1 mM EDTA and 1 mM DTT). The fractions with pure protein were pooled, concentrated using centrifugal filters, flash-frozen in liquid nitrogen and stored at −80 °C. The concentration of the protein was estimated by absorbance at 280 nm, using an extinction coefficient of 33,585 M⁻¹ cm⁻¹.

FL + 50$^{stop}$ and 1–126$^{stop}$ were generated from FL + 58$^{RNC}$ and 1–126$^{RNC}$, respectively, by replacing the SecM sequence with a stop codon using Q5 site-directed mutagenesis (NEB). Proteins were expressed as described above for RNCs. Cells were collected by centrifugation as above, and the pellet was resuspended in lysis buffer supplemented with 1 μl ml⁻¹ DNase (Roche), 1.25 mg ml⁻¹ lysozyme and 1 protease inhibitor tablet (Roche), then incubated for 30 min at

4 °C. Cells were lysed by sonication (3× 1 min, 40% amplitude) and the soluble fraction was isolated by centrifugation for 40 min at 20,000$g$ using a JA-25.50 rotor in a Beckman Avanti J-26S XP centrifuge. The supernatant was filtered (0.2 μm) and loaded onto a RESOURCE Q column connected to ÄKTA Pure Protein Purification System. The protein was eluted using a linear gradient to 50% elution buffer (25 mM Tris, pH 7.4 at 4 °C, 1 mM EDTA, 2 mM DTT and 1 M NaCl) in 20 column volumes. Fractions containing muGFP–DHFR fusion proteins were concentrated and the muGFP-tag was cleaved by incubation with 0.4 mg ml⁻¹ 3C protease overnight at 4 °C. A second, identical round of purification on a RESOURCE Q column was then performed to separate DHFR and muGFP. Fractions containing cleaved DHFR were pooled, concentrated and purified from remaining contaminants by size-exclusion chromatography using a Superdex75i column in gel filtration buffer. Fractions containing pure protein were pooled, concentrated using centrifugal filters, flash-frozen in liquid nitrogen and stored at −80 °C. Protein concentrations were determined by absorbance at 280 nm.

### Expression and purification of TF

TF was expressed with a cleavable N-terminal 6×His-tag from a pPROEX-HTa vector[58]. TF ΔRBS (F44A/R45A/K46A)[52] and monomeric TF (V39E, I76E, I80A)[60] were generated using Q5 site-directed mutagenesis (NEB).

*E. coli* BL21(DE3) cells were transformed with the TF plasmids and selected on LB agar plates supplemented with 100 μg ml⁻¹ ampicillin. A single colony was used to inoculate 1 l of ZYM 5052 autoinduction media containing 100 μg ml⁻¹ ampicillin, which was grown overnight at 37 °C. The cells were collected by centrifugation and resuspended in buffer containing 50 mM Tris-HCl, pH 7.5, 250 mM NaCl, 10% (v/v) glycerol, 2 mM β-mercaptoethanol, 0.025 units per ml benzonase and 0.1 mM PMSF. The resuspended cells were supplemented with 1 mM PMSF and a tablet of EDTA-free protease inhibitor cocktail and then lysed by sonication. The lysate was centrifuged at 48,000$g$ for 45 min at 4 °C and the supernatant was loaded onto a 5 ml HisTrap HP column connected to an ÄKTA Pure Protein Purification System. The bound protein was eluted by washing the column with a linear gradient of 0–500 mM imidazole in lysis buffer. Fractions containing pure protein were pooled, supplemented with 1:100 TEV protease to cleave the His-tag and dialyzed in a buffer containing 20 mM Tris-HCl, pH 7.5, 250 mM NaCl and 2 mM DTT at 4 °C. The protein was passed through a HisTrap column and the flow-through containing the cleaved protein was concentrated using centrifugal filters and injected into a Superdex 200i 10/300 GL column that was equilibrated with 20 mM Tris-HCl, pH 7.5, 150 mM NaCl, 5% (v/v) glycerol and 1 mM DTT. For monomeric TF, a HiPrep 26/60 Sephacryl S-300 HR column was used for the final SEC step. Fractions containing pure protein were pooled, concentrated using centrifugal filters, flash-frozen in liquid nitrogen and stored at −80 °C. The concentration of the protein was estimated using Bradford's assay[78].

### Mass spectrometry of RNCs

RNCs were purified as described above in either high-salt RNC buffer or, to preserve salt-sensitive interactions, in low-salt RNC buffer. Proteins were run for 8 mm using NuPAGE 12% Bis-Tris Gel 1.0 mm 12 wells before Coomassie blue staining (Quick Coomassie Stain, Generon). Excised entire 8 mm bands were diced, destained, alkylated and digested with trypsin (modified sequencing grade V5111, Promega). Digests were loaded onto Evotips (Evosep) and tryptic peptides were eluted using the '30SPD' gradient via an Evosep One[79] HPLC fitted with a 15-cm C18 column (EV1074) into a Lumos Tribrid Orbitrap mass spectrometer (Thermo Scientific) with a nanospray emitter operated at 2,200 V. The Orbitrap was operated in data-dependent acquisition mode with precursor ion spectra acquired at 120,000 resolution in the Orbitrap and MS/MS spectra in the ion trap at 32% HCD collision energy in TopS mode. Dynamic exclusion was set to ±10 ppm over 15 s, automatic gain

control to standard and max. injection time to dynamic. The vendor's universal method was adopted to schedule the ion trap accumulation times. Raw files were processed using Maxquant[80] (maxquant.org) and Perseus[81] (maxquant.net/perseus) with a recent download of the Uniprot *E. coli* reference proteome database together with a common contaminants database. A decoy database of reversed sequences was used to filter false positives, with both peptide and protein false detection rates set to 1%. Quantification of individual *E. coli* proteins was achieved using iBAQ (intensity-based absolute quantification) values normalized to the mean iBAQ value across all ribosomal proteins in each sample. All proteomics data have been deposited in the ProteomeXchange Consortium through the PRIDE[82] partner repository with dataset identifier PXD036784.

### Enzyme activity assays

DHFR activity was measured in low-salt RNC buffer with saturating concentrations of NADPH (Sigma-Aldrich, N6505) and DHF (Sigma-Aldrich, D7006) by following NADPH oxidation through the change in absorbance at 340 nm. For each reaction of 100 µl, 25 nM of DHFR was incubated with 100 µM NADPH for 10 min at 20 °C, then 100 µM of DHF was added and the absorbance at 340 nm was immediately recorded for 100 s. To convert the change in absorbance over time into the initial rate, the differential extinction coefficient value of 12.4 mM$^{-1}$ cm$^{-1}$ was used. All experiments were performed in triplicate at 21 °C and measured using a Jasco V-550 Spectrophotometer.

Michaelis–Menten parameters were determined by measuring enzyme activity of 25 nM DHFR as a function of increasing concentration of DHF, keeping NADPH constant at 100 µM. DHF was varied from 0.1 µM to 30 µM, and the dilutions were made in low-salt RNC buffer for all samples.

Where indicated, 500 nM methotrexate (Sigma-Aldrich, A6770) was added to the samples 20 min before measurement. Where indicated, peptides corresponding to the C terminus of DHFR were added to the samples and incubated on ice for 40 min before adding the substrates. The incubation time was optimized to ensure that the reaction had reached equilibrium. Peptides (C$^{10}$, acetyl-SYCFEILERR-amine or Scr, acetyl-RFIERCELYS-amine) were synthesized by the Peptide Chemistry Science Technology Platform (Francis Crick Institute) and reconstituted in low-salt RNC buffer.

Data were plotted and analyzed in GraphPad Prism 9.

### Pelleting assays

To measure TF binding to RNCs, 10 µM of purified TF was incubated with 1.5-2 µM RNC at 30 °C for 20 min. The reactions were then loaded onto 35% sucrose cushions prepared in either high-salt or low-salt RNC buffer and subjected to ultracentrifugation for 2 h at 250,000*g* (Beckman ultracentrifuge, TLA-100 rotor) to separate unbound TF from the ribosomal fraction. The pellet containing ribosomes was washed once, then resuspended in low-salt RNC buffer and analyzed by SDS–PAGE with Coomassie staining, or immunoblot with antibodies against TF (GenScript, A01329; 1:1,000 dilution) and small subunit ribosomal protein S2 (abx110548; 1:1,000 dilution).

### HDX MS

In addition to the descriptions below, comprehensive experimental details and parameters are provided in Supplementary Data 1, in the recommended[83] tabular format. All HDX MS data have been deposited in the ProteomeXchange Consortium through the PRIDE partner repository[82] with dataset identifier PXD036945. Supplementary Data 1 contains all the values used to create figures containing HDX MS data.

**Deuterium labeling.** Stock concentrations of purified constructs are listed in the Exp. Parameters and Replication tab in the Supplementary Data 1. All isolated proteins or RNCs began in storage buffer as follows. For free FL DHFR and DHFR RNC constructs, 50 mM HEPES-KOH, pH 7.5,

100 mM KOAc, 12 mM Mg(OAc)$_2$ and 1 mM DTT; for FL + 50$^{stop}$, 25 mM Tris, 1 mM EDTA, 1 mM DTT and 200 mM NaCl; for TF constructs, 20 mM Tris-HCL, pH 7.5, 150 mM NaCl, 5% glycerol and 1 mM DTT; for free ribosomes (obtained from NEB, P0763S), 20 mM HEPES-KOH, pH 7.6, 10 mM Mg(OAc)$_2$, 30 mM KCl and 7 mM β-mercaptoethanol. Proteins were diluted, as needed, to the concentration required for HDX and then labeled with deuterium.

Deuterium labeling was initiated with a 15-fold dilution into labeling buffer (30 µl, 10 mM HEPES-KOH, pH 7.5, 25 mM KOAc, 12 mM Mg(OAc)$_2$, 1 mM DTT and 99.9% D$_2$O). After each labeling time (10 s, 100 s and 1,000 s) at 23 °C, the labeling reaction was quenched with the addition of 20 µl of ice-cold quenching buffer (200 mM potassium phosphate, pH 2.44, 4 M guanidinium chloride, 0.72 M TCEP and H$_2$O), 10 µl 50% immobilized pepsin bead slurry (prepared in-house using POROS beads[84] in water with 0.1% formic acid and held on ice for 5 min. After on-ice in-solution digestion, the mixture was spun for 15 s at 16,000*g* and 4 °C in Corning Costar Spin-X centrifuge tube filters (Sigma-Aldrich, CLS8163-100EA) and then the flow-through was immediately injected into a Waters M-class Acquity UPLC with HDX technology for liquid chromatography–MS analysis. Undeuterated control samples were prepared for each experiment using the same procedure as outlined above but using 10 mM HEPES-KOH, pH 7.5, 25 mM KOAc, 12 mM Mg(OAc)$_2$, 1 mM DTT and 99.9% H$_2$O in place of the labeling buffer. Maximally deuterated samples (maxD) were prepared as previously described[85] for both FL DHFR and FL + 50$^{stop}$.

**Liquid chromatography–ion-mobility spectrometry–MS.** The cooling chamber of the UPLC system (based on a previous publication[86]), which housed all the chromatographic elements, was held at 0.0 ± 0.1 °C for the entire time of the measurements. Peptides were trapped and desalted on a VanGuard Pre-Column trap (2.1 mm × 5 mm; ACQUITY UPLC BEH C18, 1.7 µm; Waters, 186002346) for 3 min at 100 µl min$^{-1}$. Peptides were then eluted from the trap using a 5–35% gradient of acetonitrile over 20 min at a flow rate of 100 µl min$^{-1}$ and separated using an ACQUITY UPLC HSS T3, 1.8 µm, 1.0 mm × 50 mm column (Waters, 186003535). The back pressure averaged ~12,950 psi at 0 °C and 5% acetonitrile, 95% water, 0.1% formic acid. Mass spectra were acquired using a Waters Synapt G2-Si HDMS$^E$ mass spectrometer in ion mobility mode. The mass spectrometer was calibrated with direct infusion of a solution of glu-fibrinopeptide (Sigma-Aldrich, F3261) at 200 fmol µl$^{-1}$ at a flow rate of 5 µl min$^{-1}$ before data collection. A conventional electrospray source was used, and the instrument was scanned over a range of 50–2,000 *m/z*. The instrument configuration was as follows: capillary, 2.5 kV; trap collision energy, 4 V; sampling cone, 40 V; source temperature, 80 °C; desolvation temperature, 175 °C. All comparison experiments were performed under identical experimental conditions such that deuterium levels were not corrected for back-exchange and are therefore reported as relative[87]. Replicates were technical; that is, independent labeling reactions were performed using the same batch of purified protein. The error in determining the deuterium levels was ±0.25 Da in this experimental setup.

**HDX MS data processing.** Peptides were identified from replicate HDMS$^E$ analyses (as detailed in Supplementary Data 1) of undeuterated control samples using PLGS v.3.0.1 (Waters Corporation). Peptide masses were identified from searches using nonspecific cleavage of a custom database containing the sequences of each DHFR construct (based on wild-type *E. coli* Uniprot P0ABQ4 that included the linker sequence; see peptide maps in Supplementary Data 1), TF (*E. coli* Uniprot P0A850) and all protein sequences in the *E. coli* ribosome as extracted from PDB 4YBB. Searches used the following parameters: no missed cleavages, no PTMs, a low energy threshold of 135, an elevated energy threshold of 35 and an intensity threshold of 500. No false discovery rate control was performed. The peptides identified in PLGS (excluding all neutral loss and in-source fragmentation identifications)

were then filtered in DynamX v.3.0 (Waters Corporation), implementing minimum products per amino acid and consecutive product ion cutoffs described in the Exp. Parameters and Replication tab of Supplementary Data 1. Peptides meeting the filtering criteria to this point were further processed by DynamX v.3.0 (Waters Corporation), including manual inspection of each mass spectrum. The relative amount of deuterium in each peptide was determined with the software by subtracting the centroid mass of the undeuterated form of each peptide from the deuterated form, at each time point, for each condition. These deuterium uptake values were used to generate all uptake graphs and difference maps.

### Reporting summary

Further information on research design is available in the Nature Portfolio Reporting Summary linked to this article.

### Data availability

All mass spectrometry data have been deposited in the ProteomeXchange Consortium through the PRIDE partner repository with dataset identifiers PXD036784 and PXD036945. Proteomic analysis used the Uniprot *E. coli* reference proteome (UP000000625). Source data are provided with this paper.

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

### Acknowledgements

We thank J. Christodoulou (University College London) for the kind gift of the Δ*tig E. coli* strain; S. Mouilleron (Francis Crick Institute) for the 3C protease and TEV protease; K. Fadgen (Waters) for technical assistance with HDX MS; and the Peptide Chemistry Science Technology Platform at the Francis Crick institute for peptide synthesis. D.B. is supported by the Francis Crick Institute, which receives its core funding from Cancer Research UK (CC2025, CC1063, CC1065 and CC1068), the UK Medical Research Council (CC2025, CC1063, CC1065 and CC1068) and the Wellcome Trust (CC2025, CC1063, CC1065 and CC1068). For the purpose of open access, the author has applied a CC BY public copyright license to any author accepted manuscript version arising from this submission. J.R.E. acknowledges funding from the National Institutes of Health (R01-CA233978) and the James L. Waters Chair in Analytical Chemistry.

### Author contributions

F.U.H., J.R.E. and D.B. conceived the project. T.E.W., A.P., A.R., S.S., S.H., S.K. and D.B. conducted the investigation. T.E.W., A.P., A.R., S.S., S.H., J.R.E. and D.B. performed the formal analysis, data curation and visualization: T.E.W., A.P. and D.B. wrote the original draft of the manuscript; all authors reviewed and edited the final draft.

### Funding

### Competing interests

The authors declare no competing interests.

### Additional information

**Extended data** is available for this paper at https://doi.org/10.1038/s41594-024-01355-x.

**Correspondence and requests for materials** should be addressed to John R. Engen or David Balchin.

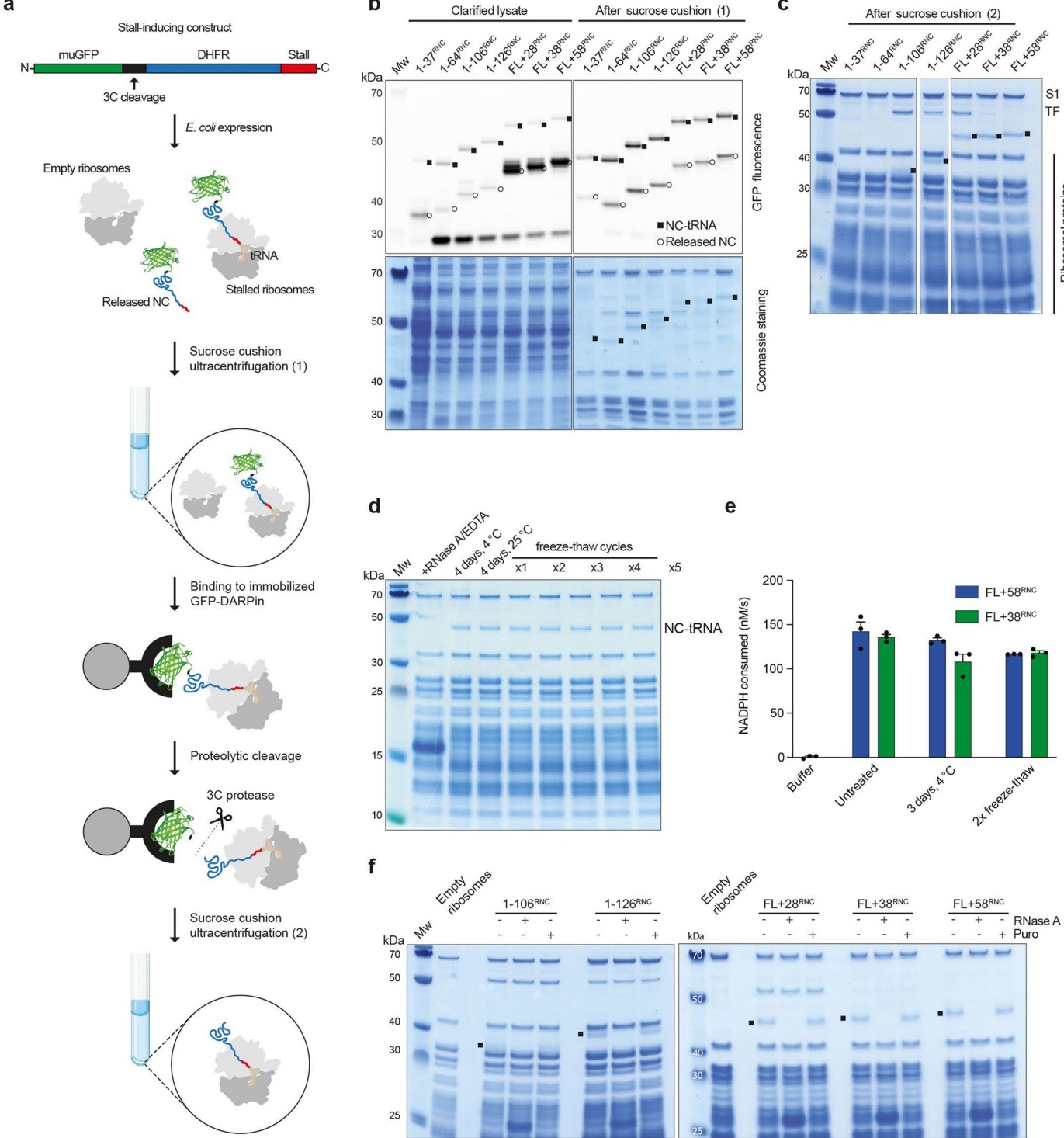

**Extended Data Fig. 1 | See next page for caption.**

**Extended Data Fig. 1 | Preparation and quality control of stalled ribosome:nascent chain complexes (RNCs). a**, RNC purification. Constructs encode an N-terminal muGFP followed by a cleavage site for 3 C protease, DHFR sequence, and a C-terminal stalling sequence derived from *M. succiniciproducens* SecM. Expression in *E. coli* produces both stalled RNCs and free NCs. Total ribosomes are pelleted by sucrose cushion centrifugation, then RNCs are purified using a GFP-binding DARPin. RNCs are selectively eluted using 3 C protease, leaving GFP on the resin. A second ultracentrifugation step removes residual free NC. **b**, RNCs are tracked by SDS-PAGE followed by either fluorescent imaging of the GFP or Coomassie staining. Ribosome-bound NCs are covalently coupled to peptidyl tRNA ( ~ 20 kDa) and can therefore be distinguished from released NCs. Note that shortest RNC (1-37$^{RNC}$) contains an additional disordered linker after the GFP to improve capture by the DARPin. It therefore migrates slower on SDS-PAGE than 1-64$^{RNC}$. Residual released NC, detectable by sensitive fluorescence imaging, is removed by the second round of centrifugation. Experiments were repeated 5 times with similar results. **c**, SDS-PAGE with Coomassie staining of purified RNCs. Ribosomal proteins including S1 are indicated, as is Trigger factor. Where visible, the NC-tRNA band is indicated. Experiments were repeated 5 times with similar results. **d**, RNCs are stable over prolonged incubation and multiple freeze-thaw cycles. The stability of FL + 58$^{RNC}$ was monitored by the integrity of the NC-tRNA band on SDS-PAGE. Conditions included incubation for 4 days at 25 °C or 4 °C, or repeated cycles of freezing in liquid N$_2$ followed by thawing on ice. As a positive control, the RNC was disrupted by treatment with 50 µg/mL RNaseA and 50 mM EDTA. **e**, DHFR in full-length RNCs is stable over prolonged incubation and multiple freeze-thaw cycles. Enzyme activity of FL + 38$^{RNC}$ and FL + 58$^{RNC}$ was measured either immediately after purification, after incubation at 4 °C for 3 days, or after two freeze-thaw cycles as described in **d**. Data are presented as mean values ± SD, n = 3 independent experiments. **f**, Purified RNCs were treated with either 2.5 mM puromycin or 50 µg/mL RNaseA. The NC-tRNA band is indicated in the untreated control lanes. Experiments were repeated twice with similar results.

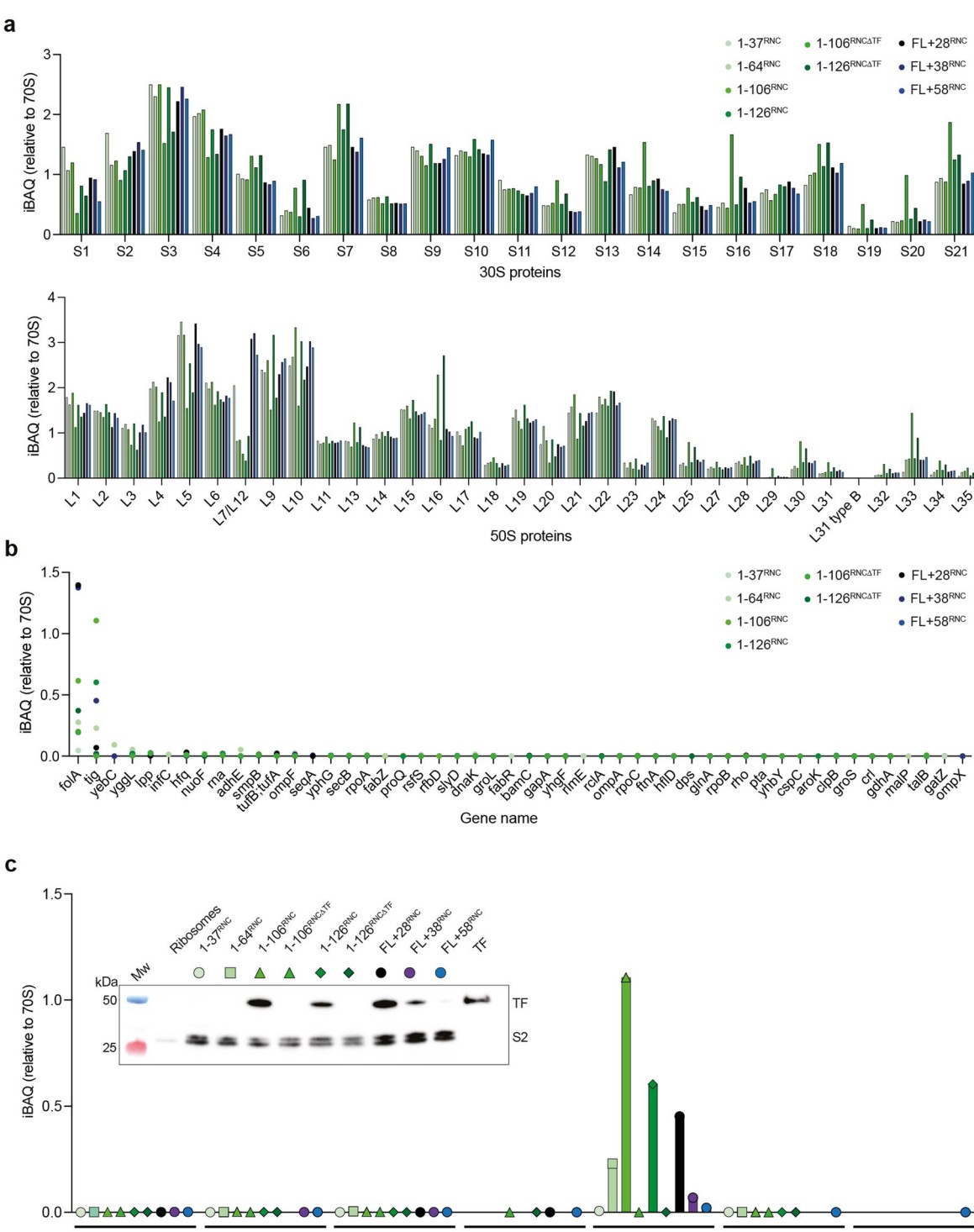

**Extended Data Fig. 2 | Mass spectrometric analysis of RNC composition.**
**a**, Small (30 S) and large (50 S) subunit ribosomal proteins detected in purified RNCs. Abundances are based on iBAQ values and normalized to the average iBAQ across all ribosomal proteins in that RNC sample, which is set as 1. **b**, Fifty most abundant proteins in RNCs, not including ribosomal proteins. Interactor stoichiometry is calculated as in **a**, with the average iBAQ of all detected ribosomal proteins set to 1. iBAQ values for DHFR (*folA*) were not corrected for NC length. **c**, Chaperones detected by MS analysis of RNCs. Data are normalized as in **b**. The inset shows a western blot of the purified RNCs, with antibodies directed against Trigger factor or 30 S protein S2 as a loading control. Western blot experiments were repeated three times with similar results.

**a**

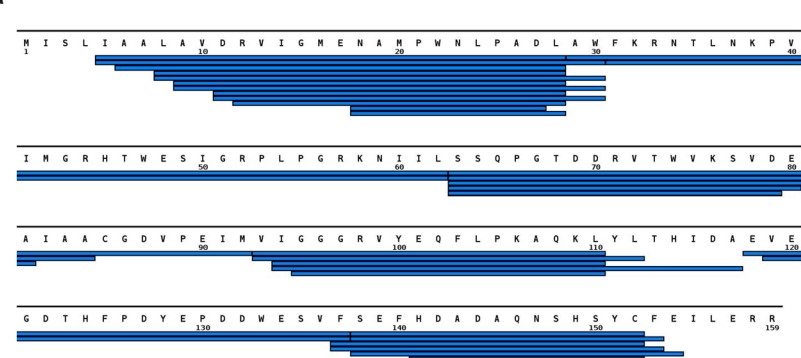

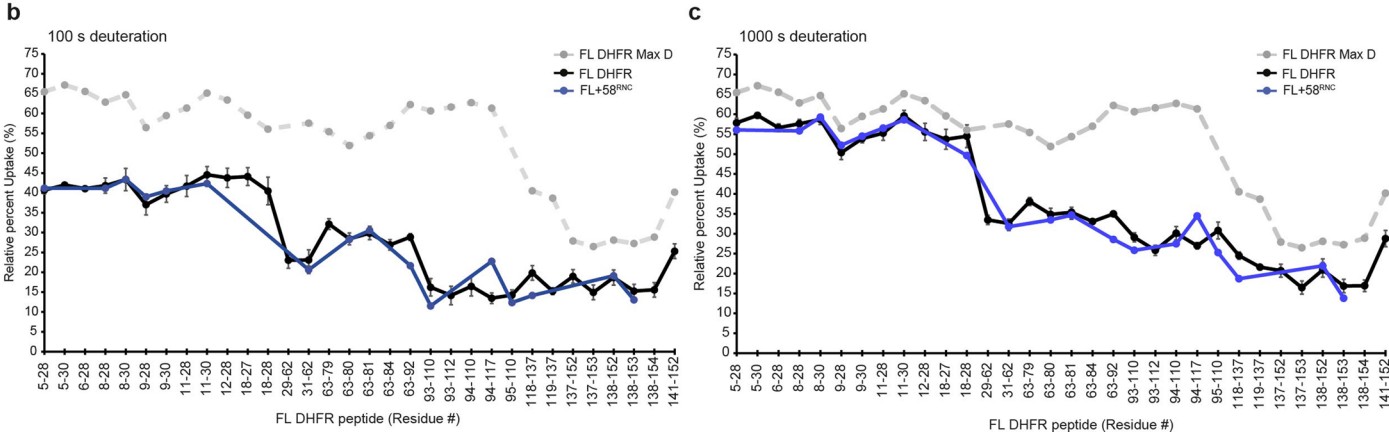

**Extended Data Fig. 3 | HDX MS analysis of full-length DHFR on the ribosome.**
**a**, Peptide sequence coverage of DHFR in FL + 58$^{RNC}$ (34 peptides, 94% coverage).
Each peptide is represented by a blue bar. **b**, Relative deuterium uptake of DHFR
peptides after 100 s exposure to deuterium, as a percentage of the maximum
possible exchange, for isolated DHFR (FL DHFR) and FL + 58$^{RNC}$. A maximally-
deuterated control sample (FL DHFR Max D) is shown as a reference. Data are
presented as mean values of 2-4 replicates. Where shown, error bars represent
SD, n = 3 or n = 4 independent experiments. See also Data S1. **c**, As in panel **b**, but
showing relative deuterium uptake of DHFR peptides after 1000 s exposure to
deuterium.

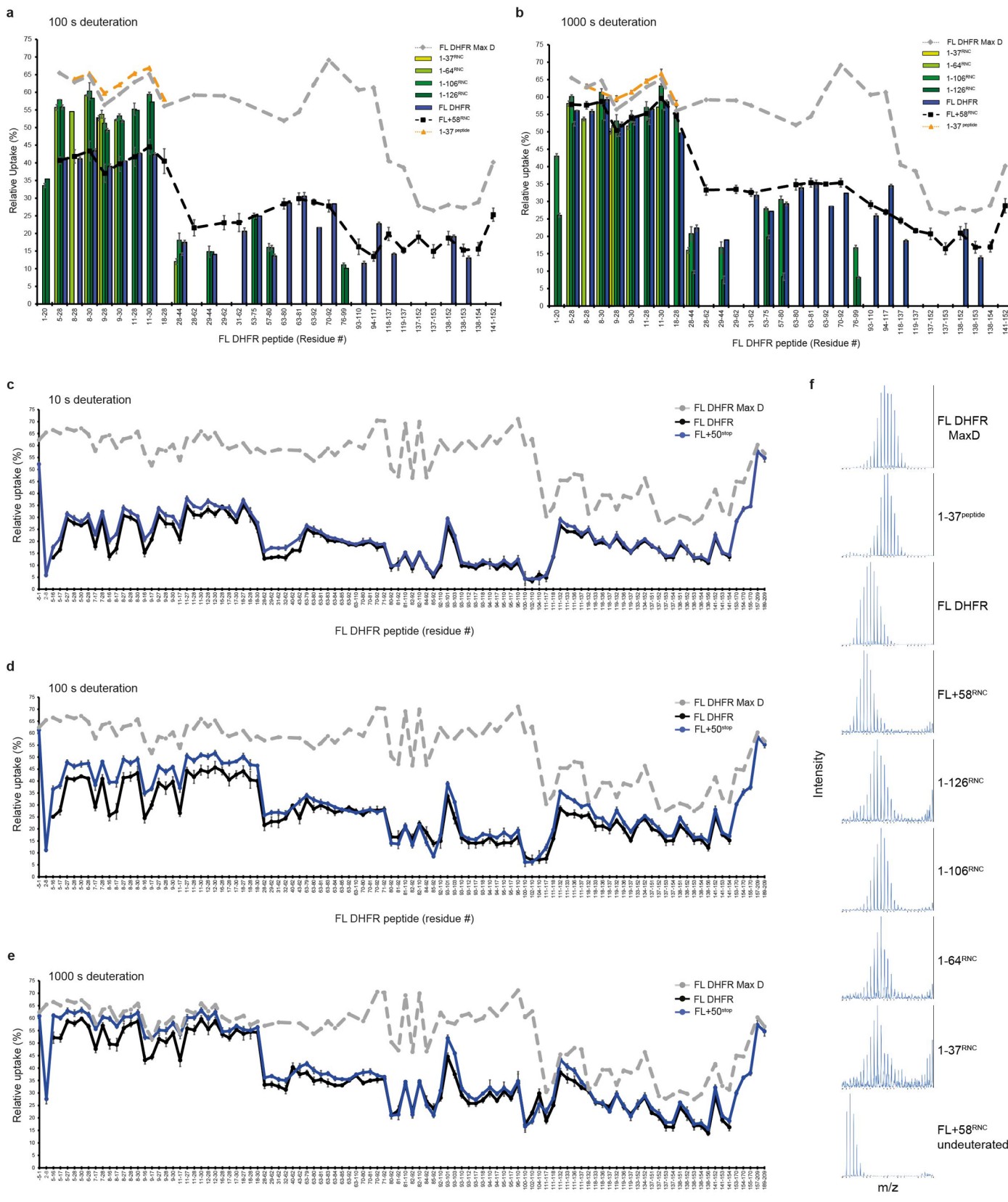

**Extended Data Fig. 4 | HDX MS analysis of DHFR RNCs. a**, Relative deuterium uptake of DHFR peptides after 100 s exposure to deuterium. Values are the average of 2-4 replicates. Error bars represent s.d. See also Data S1. **b**, As in panel **a**, but showing relative deuterium uptake of DHFR peptides after 1000 s exposure to deuterium. **c**, Relative deuterium uptake of DHFR peptides after 10 s exposure to deuterium, as a percentage of the maximum possible exchange, for isolated DHFR (FL DHFR) and FL+50^stop. A maximally-deuterated control sample (FL+50^stop Max D) is shown as a reference. Data are presented as mean values of 2-4 replicates. Where shown, error bars represent SD, n = 3 or n = 4 independent experiments. **d**, As in panel **c**, but for 100 s exposure to deuterium. **e**, As in panel **c**, but for 1000 s exposure to deuterium. **f**, Representative mass spectra for peptide 9-28, charge state +2, after 10 s deuteration. See also Data S1.

a

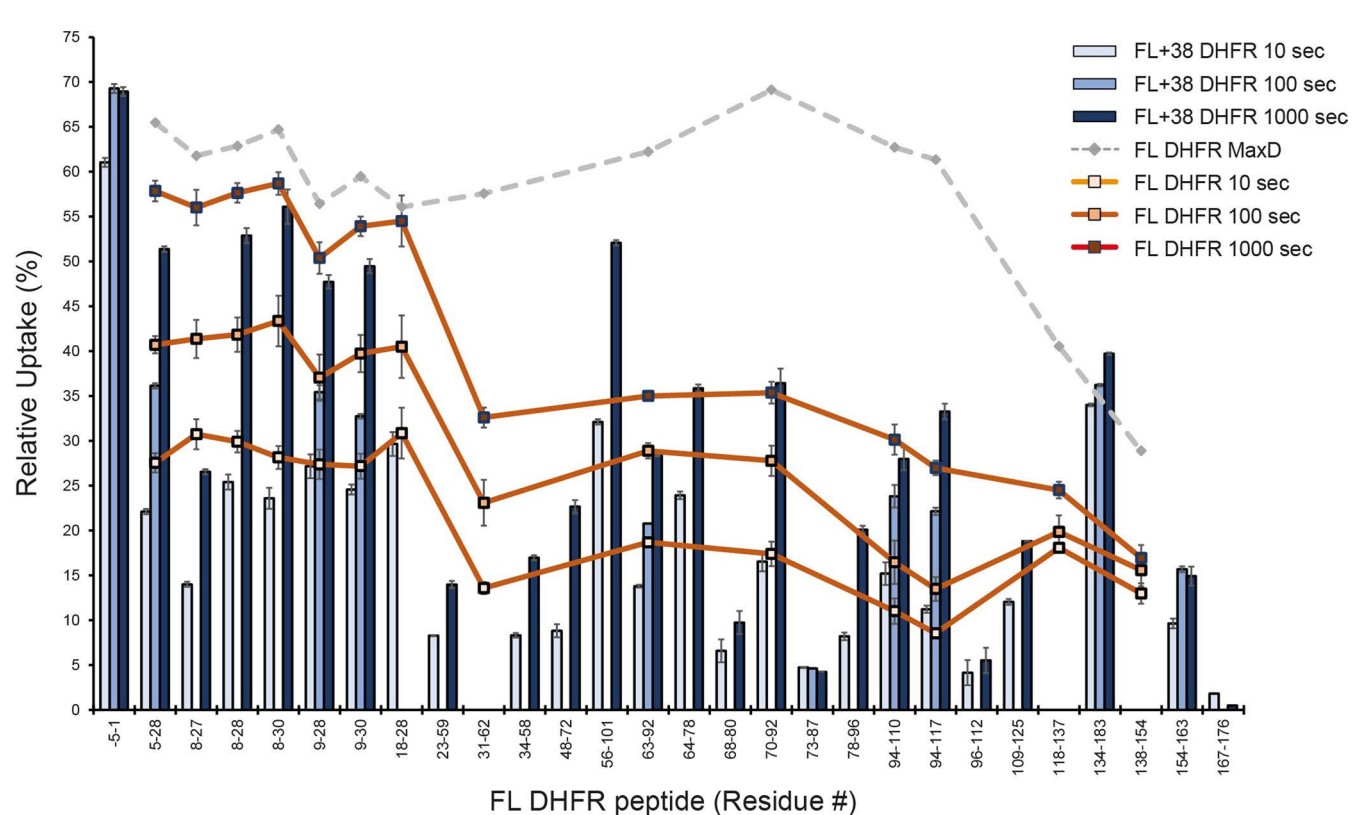

**Extended Data Fig. 5 | HDX MS analysis of FL + 38<sup>RNC</sup>. a**, Relative deuterium uptake of DHFR peptides after 10 s, 100 s or 1000 s exposure to deuterium, as a percentage of the maximum possible exchange, for isolated DHFR (FL DHFR) and FL + 38$^{RNC}$. A maximally-deuterated control sample (FL DHFR Max D) is shown as a reference. Data are presented as mean values of 2-4 replicates. Where shown, error bars represent SD, n = 3 or n = 4 independent experiments. See also Data S1.

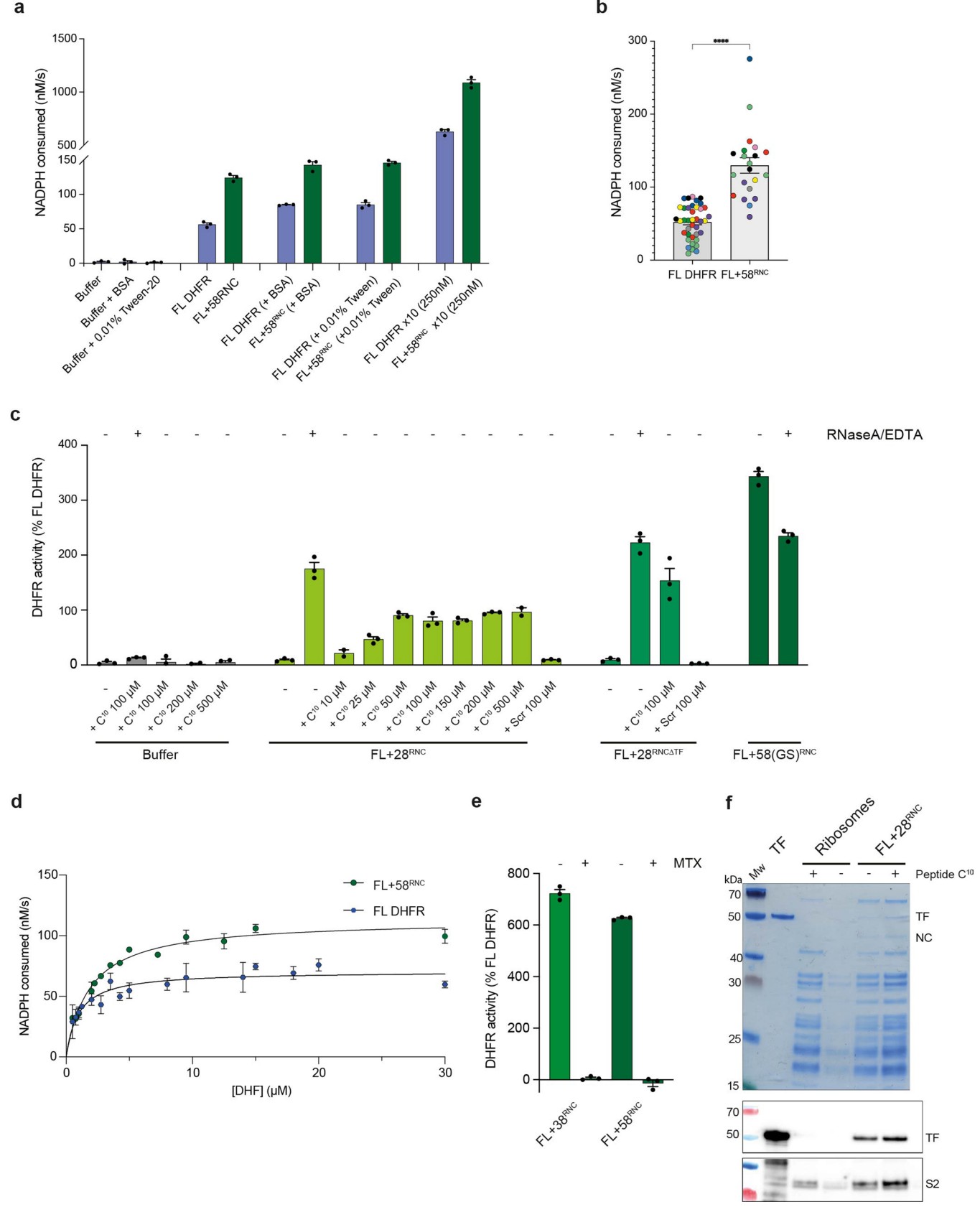

**Extended Data Fig. 6 | See next page for caption.**

**Extended Data Fig. 6 | Modulation of DHFR activity on the ribosome.**
**a**, Enzyme activity of FL DHFR and FL + 58$^{RNC}$, measured as in Fig. 4b, at a
concentration of 25 or 250 nM, supplemented with either 1 mg/ml BSA or 0.01%
tween20. **b**, Replicate activity measurements for FL DHFR and FL + 58$^{RNC}$.
Data are shown for 3 independent purifications of FL DHFR and 11 independent
purifications of FL + 58$^{RNC}$. Each point is the average of three technical replicates.
Error bars represent SD. Samples measured on the same day, using the same
batch of NADPH and DHF, are shown in the same colour. The activity of FL DHFR
and FL + 58$^{RNC}$ is significantly different with a two-tailed p-value = 1×10$^{-11}$ (unequal
variance Welch's t-test). **c**, Oxidoreductase activity of 25 nM FL DHFR or RNCs,
normalized to FL DHFR. Where indicated, reactions were supplemented with
50 µg/ml RNaseA and 50 mM EDTA, a peptide corresponding to the C-terminus
of DHFR (C$^{10}$, SYCFEILERR), or a scrambled-sequence control peptide (Scr,
RFIERCELYS). FL + 58(GS)$^{RNC}$ is a version of FL + 58$^{RNC}$, with the linker between
DHFR and the stalling sequence replaced by 25xGS repeats. Data are presented

as mean values ± SD, n = 3 independent experiments. **d**, Activity of 25 nM FL
DHFR or FL + 58$^{RNC}$, at different concentrations of dihydrofolate (DHF). For
FL DHFR, K$_M$ = 1.0 µM (95% CI: 0.7 to 1.3); V$_{max}$ = 71 nM/s (95% CI: 66 to 75). For
FL + 58$^{RNC}$, K$_M$ = 1.9 µM (95% CI: 1.6 to 2.2); V$_{max}$ = 113 nM/s (95% CI: 108 to 118).
Data are presented as mean values ± SD, n = 3 independent experiments. **e**,
DHFR RNC activity is sensitive to methotrexate. Activity of 25 nM FL + 38$^{RNC}$ or
FL + 58$^{RNC}$, with or without 500 nM methotrexate (MTX). Activity is normalized
to untreated FL DHFR. Data are presented as mean values ± SD, n = 3 independent
experiments. **f**, Pelleting assay showing TF binding to complemented FL + 28$^{RNC}$.
Empty ribosomes or FL + 28$^{RNC}$ were mixed with 100 µM peptide C$^{10}$ and
centrifuged through a high-salt sucrose cushion. Resuspended pellets were
analyzed by SDS-PAGE with Coomassie staining, and immunoblot against TF, with
S2 was a loading control. Purified TF, not subjected to centrifugation, is shown for
reference. Experiments were repeated three times with similar results.

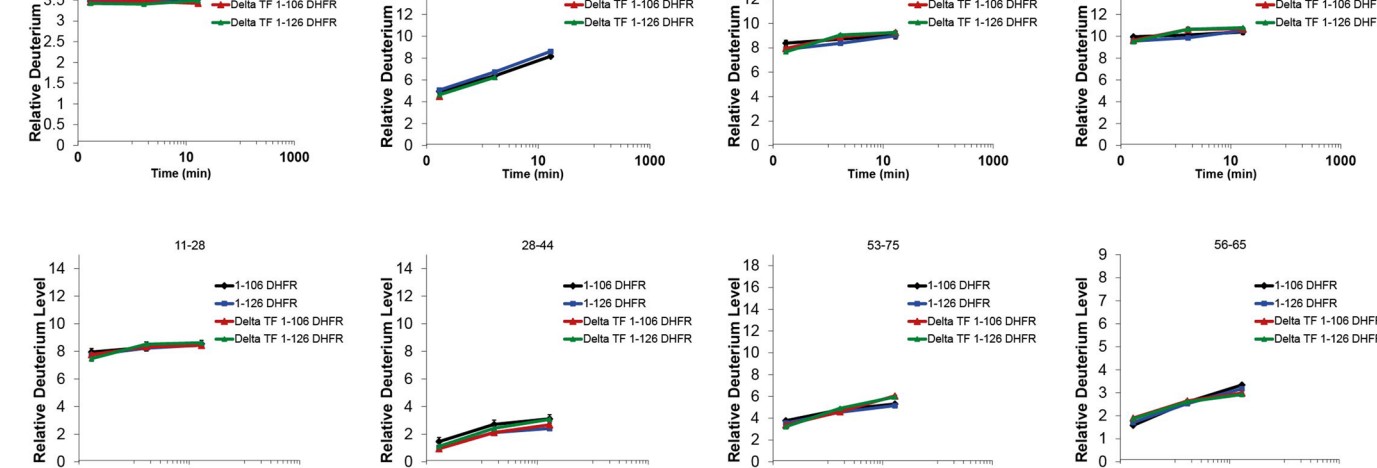

**a**, Peptide MQVSVETTQGLGRRVTITIAADSIETAVKSELVNVAKKVRIDGFRKGKVPMNIVAQRYGASVRQDVLGDLMSRNFIDAIIKEKINPAGAPTYVPGEYKLGEDFTYSVEFEVYPEVELQGLEAIEVEKPIVEVTDADVDGMLDTLRKQQAT

WKEKDGAVEAEDRVTIDFTGSVDGEEFEGGKASDFVLAMGQGRMIPGFEDGIKGHKAGEEFTIDVTFPPEEYHAENLKGKAAKFAINLKKVEERELPELTAEFIKRFGVEDGSVEGLRAEVRKNMERELKSAIRNRVKSQAIEGLVKANDI

DVPAALIDSEIDVLRRQAAQRFGGNEKQALELPRELFEEQAKRRVVVGLLLGEVIRTNELKADEERVKGLIEEMASAYEDPKEVIEFYSKNKELMDNMRNVALEEQAVEAVLAKAKVTEKETTFNELMNQQA

**Extended Data Fig. 7 | HDX MS analysis of TF and ΔTF RNCs. a**, Peptide sequence coverage of TF in RNCs (181 peptides, 99.5% coverage). Each peptide is represented by a blue bar. **b**, Structure of TF dimer (PDB: 6D6S), with peptides that are deprotected in monomeric TF relative to wild-type TF, at any deuteration time point, colored red. One monomer is shown in surface representation. See also Data S1. **c**, Relative deuterium uptake as a function of deuteration time for selected peptides in 1-106[RNC] and 1-126[RNC], with and without TF. Data are presented as mean values ± SD, n = 3 independent experiments. See also Data S1.

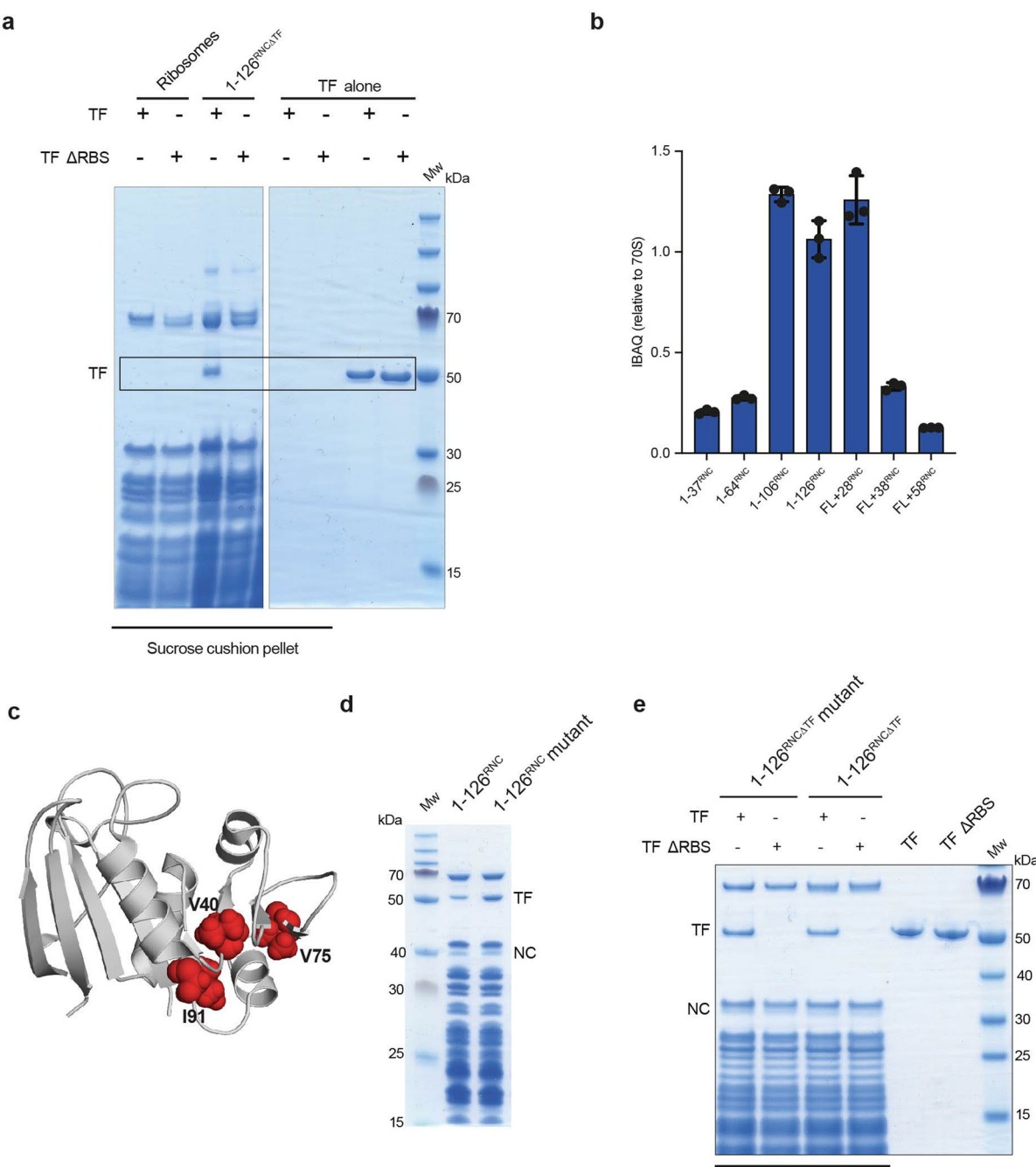

**Extended Data Fig. 8 | Determinants of Trigger factor (TF) binding to RNCs. a**, Empty 70 S ribosomes or 1-126$^{RNC\Delta TF}$ were incubated with wild-type or ribosome-binding-impaired (ΔRBS, F44A/R45A/K46A) TF and the reactions were centrifuged through a high-salt (1 M KOAc) sucrose cushion. The pellets were resuspended and analyzed by SDS-PAGE. No TF pelleted in the absence of ribosomes or RNCs. Experiments were repeated twice with similar results. **b**, Quantitative proteomic analysis of TF occupancy (as in Extended Data Fig. 2c) on RNCs purified under low salt (100 mM KOAc) conditions. Data are presented

as mean values ± SD, n = 3 independent experiments. **c**, Structure of DHFR, with the residues mutated in the destabilized variant shown in red. **d**, SDS-PAGE of wild-type 1-126$^{RNC}$ and destabilized mutant (1-126$^{RNC}$ mutant) purified under high salt. Experiments were repeated twice with similar results. **e**, TF binding to wild-type 1-126$^{RNC\Delta TF}$ and destabilized mutant (1-126$^{RNC\Delta TF}$ mutant) was analyzed as in **a**, except that the reactions were centrifuged through a low-salt sucrose cushion. Experiments were repeated twice with similar results.

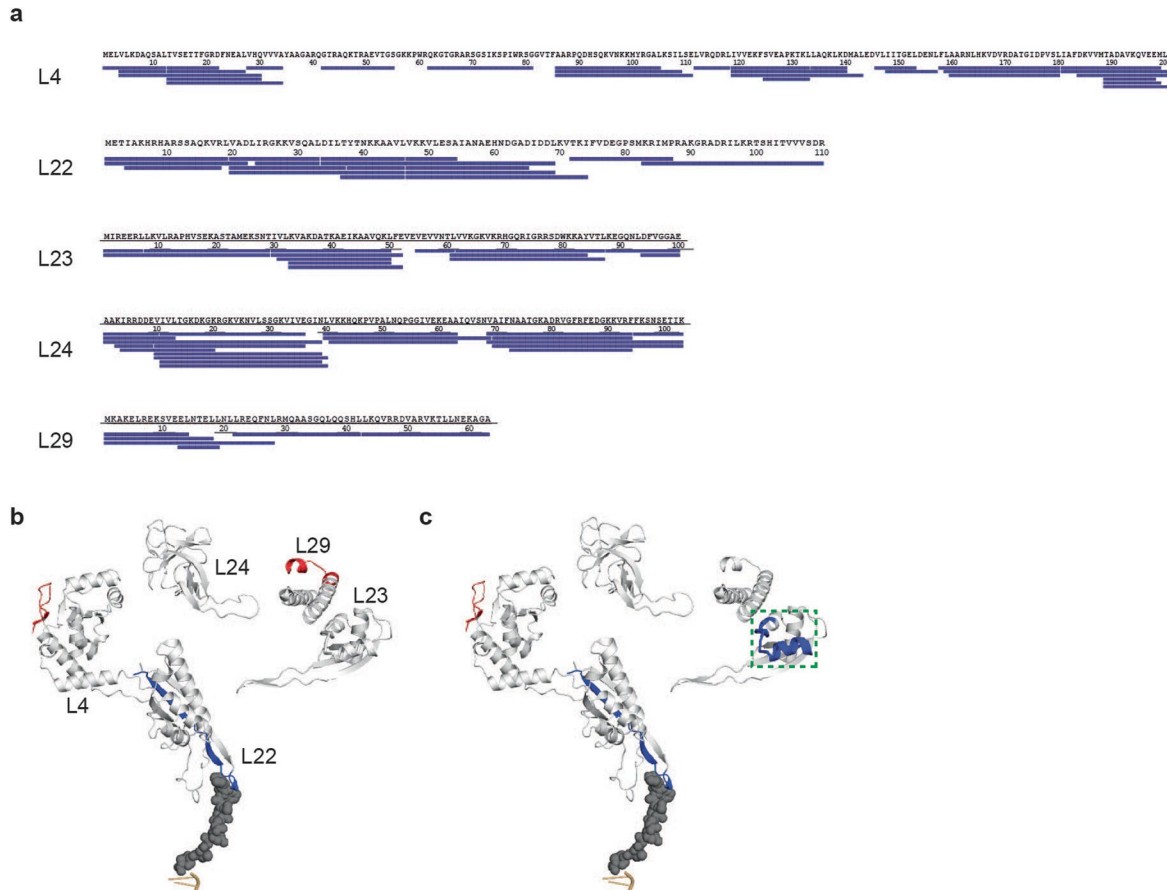

**Extended Data Fig. 9 | HDX MS analysis of ribosomal proteins. a**, Peptide sequence coverage of ribosomal proteins L4 (30 peptides, 90.5% coverage), L22 (17 peptides, 100% coverage), L23 (14 peptides, 98% coverage), L24 (20 peptides, 100% coverage), and L29 (6 peptides, 100% coverage). **b**, HDX MS analysis of riboso proteins in 1-106RNCΔTF. Peptides that are protected from HDX relative to empty ribosomes, at any deuteration time point, are colored blue. Deprotected peptides are colored red. **c**, HDX MS analysis of ribosomal proteins in 1-126RNCΔTF, as in (B). The TF-docking site on L23 is boxed. See also Data S1.

**Extended Data Table 1 | DHFR RNC sequences**

| Name | DHFR length (aa) | Amino acid sequence |
|------|------------------|---------------------|
| **1-37<sup>RNC</sup>** | 1-37 | MSKGEELFTGVVPILVELDGDVNGHKFSVRGEGEGDATNGKLTLKFICTTGKLPVPWPTLVTTLTYGVL CFSRYPDHMKRHDFFKSAMPEGYVQERTISFKDDGTYKTRAEVKFEGDTLVNRIELKGIDFKEDGNILG HKLEYNFNSHNVYITADKQKNGIKAYFKIRHNVEDGSVQLADHYQQNTPIGDGPVLLPDNHYLSTQSVL SKDPNEKRDHMVLLEDVTAAGITHGMDELYK**GSGS**LEVLFQGP**GSGSGSGSGSGSGSGSGSGSGSGSGS ENLYFQGSG**MISLIAALAVDRVIGMENAMPWNLPADLAWFKRNTLNWWWPRIRGPPGS |
| **1-64<sup>RNC</sup>** | 1-64 | MSKGEELFTGVVPILVELDGDVNGHKFSVRGEGEGDATNGKLTLKFICTTGKLPVPWPTLVTTLTYGVL CFSRYPDHMKRHDFFKSAMPEGYVQERTISFKDDGTYKTRAEVKFEGDTLVNRIELKGIDFKEDGNILG HKLEYNFNSHNVYITADKQKNGIKAYFKIRHNVEDGSVQLADHYQQNTPIGDGPVLLPDNHYLSTQSVL SKDPNEKRDHMVLLEDVTAAGITHGMDELYK**GSGS**LEVLFQGP**GSGS**MISLIAALAVDRVIGMENAMPW NLPADLAWFKRNTLNKPVIMGRHTWESIGRPLPGRKNIILSSWWWPRIRGPPGS |
| **1-106<sup>RNC</sup>** | 1-106 | MSKGEELFTGVVPILVELDGDVNGHKFSVRGEGEGDATNGKLTLKFICTTGKLPVPWPTLVTTLTYGVL CFSRYPDHMKRHDFFKSAMPEGYVQERTISFKDDGTYKTRAEVKFEGDTLVNRIELKGIDFKEDGNILG HKLEYNFNSHNVYITADKQKNGIKAYFKIRHNVEDGSVQLADHYQQNTPIGDGPVLLPDNHYLSTQSVL SKDPNEKRDHMVLLEDVTAAGITHGMDELYK**GSGS**LEVLFQGP**GSGS**MISLIAALAVDRVIGMENAMPW NLPADLAWFKRNTLNKPVIMGRHTWESIGRPLPGRKNIILSSQPGTDDRVTWVKSVDEAIAACGDVPEI MVIGGGRVYEQFLPKWWWWPRIRGPPGS |
| **1-126<sup>RNC</sup>** | 1-126 | MSKGEELFTGVVPILVELDGDVNGHKFSVRGEGEGDATNGKLTLKFICTTGKLPVPWPTLVTTLTYGVL CFSRYPDHMKRHDFFKSAMPEGYVQERTISFKDDGTYKTRAEVKFEGDTLVNRIELKGIDFKEDGNILG HKLEYNFNSHNVYITADKQKNGIKAYFKIRHNVEDGSVQLADHYQQNTPIGDGPVLLPDNHYLSTQSVL SKDPNEKRDHMVLLEDVTAAGITHGMDELYK**GSGS**LEVLFQGP**GSGS**MISLIAALAVDRVIGMENAMPW NLPADLAWFKRNTLNKPVIMGRHTWESIGRPLPGRKNIILSSQPGTDDRVTWVKSVDEAIAACGDVPEI MVIGGGRVYEQFLPKAQKLYLTHIDAEVEGDTHFPWWWPRIRGPPGS |
| **1-126<sup>RNC</sup> mutant (V40A, V75H, I91L)** | 1-126 | MSKGEELFTGVVPILVELDGDVNGHKFSVRGEGEGDATNGKLTLKFICTTGKLPVPWPTLVTTLTYGVL CFSRYPDHMKRHDFFKSAMPEGYVQERTISFKDDGTYKTRAEVKFEGDTLVNRIELKGIDFKEDGNILG HKLEYNFNSHNVYITADKQKNGIKAYFKIRHNVEDGSVQLADHYQQNTPIGDGPVLLPDNHYLSTQSVL SKDPNEKRDHMVLLEDVTAAGITHGMDELYK**GSGS**LEVLFQGP**GSGS**MISLIAALAVDRVIGMENAMPW NLPADLAWFKRNTLNKP<u>A</u>IMGRHTWESIGRPLPGRKNIILSSQPGTDDRVT<u>H</u>KSVDEAIAACGDVPE<u>L</u> MVIGGGRVYEQFLPKAQKLYLTHIDAEVEGDTHFPWWWPRIRGPPGS |
| **FL+28<sup>RNC</sup>** | 1-159 | MSKGEELFTGVVPILVELDGDVNGHKFSVRGEGEGDATNGKLTLKFICTTGKLPVPWPTLVTTLTYGVL CFSRYPDHMKRHDFFKSAMPEGYVQERTISFKDDGTYKTRAEVKFEGDTLVNRIELKGIDFKEDGNILG HKLEYNFNSHNVYITADKQKNGIKAYFKIRHNVEDGSVQLADHYQQNTPIGDGPVLLPDNHYLSTQSVL SKDPNEKRDHMVLLEDVTAAGITHGMDELYK**GSGS**LEVLFQGP**GSGS**MISLIAALAVDRVIGMENAMPW NLPADLAWFKRNTLNKPVIMGRHTWESIGRPLPGRKNIILSSQPGTDDRVTWVKSVDEAIAACGDVPEI MVIGGGRVYEQFLPKAQKLYLTHIDAEVEGDTHFPDYEPDDWESVFSEFHDADAQNSHSYCFEILERR**G SGGDLGSGGYLGSGGLDGSGGVEGSGGFL**WWWPRIRGPPGS |
| **FL+38<sup>RNC</sup>** | 1-159 | MSKGEELFTGVVPILVELDGDVNGHKFSVRGEGEGDATNGKLTLKFICTTGKLPVPWPTLVTTLTYGVL CFSRYPDHMKRHDFFKSAMPEGYVQERTISFKDDGTYKTRAEVKFEGDTLVNRIELKGIDFKEDGNILG HKLEYNFNSHNVYITADKQKNGIKAYFKIRHNVEDGSVQLADHYQQNTPIGDGPVLLPDNHYLSTQSVL SKDPNEKRDHMVLLEDVTAAGITHGMDELYK**GSGS**LEVLFQGP**GSGS**MISLIAALAVDRVIGMENAMPW NLPADLAWFKRNTLNKPVIMGRHTWESIGRPLPGRKNIILSSQPGTDDRVTWVKSVDEAIAACGDVPEI MVIGGGRVYEQFLPKAQKLYLTHIDAEVEGDTHFPDYEPDDWESVFSEFHDADAQNSHSYCFEILERRG SGGDLGSGGYLGSGGLDGSGGVEGSGGFLWWWPRIRGPPGS |
| **FL+58<sup>RNC</sup>** | 1-159 | MSKGEELFTGVVPILVELDGDVNGHKFSVRGEGEGDATNGKLTLKFICTTGKLPVPWPTLVTTLTYGVL CFSRYPDHMKRHDFFKSAMPEGYVQERTISFKDDGTYKTRAEVKFEGDTLVNRIELKGIDFKEDGNILG HKLEYNFNSHNVYITADKQKNGIKAYFKIRHNVEDGSVQLADHYQQNTPIGDGPVLLPDNHYLSTQSVL SKDPNEKRDHMVLLEDVTAAGITHGMDELYK**GSGS**LEVLFQGP**GSGS**MISLIAALAVDRVIGMENAMPW NLPADLAWFKRNTLNKPVIMGRHTWESIGRPLPGRKNIILSSQPGTDDRVTWVKSVDEAIAACGDVPEI MVIGGGRVYEQFLPKAQKLYLTHIDAEVEGDTHFPDYEPDDWESVFSEFHDADAQNSHSYCFEILERR**G SGGDLGSGGYLGSGGLDGSGGVEGSGGFLGSGGLDGSGGLYGSGGVEGS**WWWPRIRGPPGS |
| **FL+58(GS)<sup>RNC</sup>** | 1-159 | MSKGEELFTGVVPILVELDGDVNGHKFSVRGEGEGDATNGKLTLKFICTTGKLPVPWPTLVTTLTYGVL CFSRYPDHMKRHDFFKSAMPEGYVQERTISFKDDGTYKTRAEVKFEGDTLVNRIELKGIDFKEDGNILG HKLEYNFNSHNVYITADKQKNGIKAYFKIRHNVEDGSVQLADHYQQNTPIGDGPVLLPDNHYLSTQSVL SKDPNEKRDHMVLLEDVTAAGITHGMDELYK**GSGS**LEVLFQGP**GSGS**MISLIAALAVDRVIGMENAMPW NLPADLAWFKRNTLNKPVIMGRHTWESIGRPLPGRKNIILSSQPGTDDRVTWVKSVDEAIAACGDVPEI MVIGGGRVYEQFLPKAQKLYLTHIDAEVEGDTHFPDYEPDDWESVFSEFHDADAQNSHSYCFEILERR**G SGSGSGSGSGSGSGSGSGSGSGSGSGSGSGSGSGSGSGSGSGSGSGSGSGS**WWWPRIRGPPGS |

The N-terminal muGFP (green), functioning as an affinity tag, precedes a 3C cleavage sequence (blue) that is flanked on either side by (GS)2 linkers (bold). The subsequent DHFR sequence fragment is followed by a C-terminal stall-inducing sequence derived from *M. succiniciproducens* SecM. When describing our RNCs we assume that the ribosome stalls with the penultimate tRNAPro in the A-site and the first 8 residues from the stall-inducing sequence occupying the exit tunnel. In the full-length RNCs an additional disordered linker precedes the stall sequence (bold). In 1-37RNC, the GS linker following the 3C cleavage sequence is extended to improve capture by the GFP affinity resin. Sites of point mutations are underlined.

# Reporting Summary

## Statistics

For all statistical analyses, confirm that the following items are present in the figure legend, table legend, main text, or Methods section.

| n/a | Confirmed | |
|---|---|---|
| ☐ | ☒ | The exact sample size (*n*) for each experimental group/condition, given as a discrete number and unit of measurement |
| ☐ | ☒ | A statement on whether measurements were taken from distinct samples or whether the same sample was measured repeatedly |
| ☐ | ☒ | The statistical test(s) used AND whether they are one- or two-sided<br>*Only common tests should be described solely by name; describe more complex techniques in the Methods section.* |
| ☒ | ☐ | A description of all covariates tested |
| ☒ | ☐ | A description of any assumptions or corrections, such as tests of normality and adjustment for multiple comparisons |
| ☐ | ☒ | A full description of the statistical parameters including central tendency (e.g. means) or other basic estimates (e.g. regression coefficient) AND variation (e.g. standard deviation) or associated estimates of uncertainty (e.g. confidence intervals) |
| ☐ | ☒ | For null hypothesis testing, the test statistic (e.g. *F*, *t*, *r*) with confidence intervals, effect sizes, degrees of freedom and *P* value noted<br>*Give P values as exact values whenever suitable.* |
| ☒ | ☐ | For Bayesian analysis, information on the choice of priors and Markov chain Monte Carlo settings |
| ☒ | ☐ | For hierarchical and complex designs, identification of the appropriate level for tests and full reporting of outcomes |
| ☒ | ☐ | Estimates of effect sizes (e.g. Cohen's *d*, Pearson's *r*), indicating how they were calculated |

*Our web collection on statistics for biologists contains articles on many of the points above.*

## Software and code

Policy information about availability of computer code

| | |
|---|---|
| Data collection | HDX-MS data were collected using a Synapt G2Si (Waters). Proteomics data were collected using a Lumos Tribrid Orbitrap. |
| Data analysis | PLGS 3.0 and Dynamx 3.0 (Waters) were used for analysis of HDX-MS data. Proteomics data were analysed using Maxquant 2.5 and Perseus 1.6.2.3. Other data were analyzed using Graphpad Prism 9. |

For manuscripts utilizing custom algorithms or software that are central to the research but not yet described in published literature, software must be made available to editors and reviewers. We strongly encourage code deposition in a community repository (e.g. GitHub). See the Nature Portfolio guidelines for submitting code & software for further information.

## Data

Policy information about availability of data

All manuscripts must include a data availability statement. This statement should provide the following information, where applicable:
- Accession codes, unique identifiers, or web links for publicly available datasets
- A description of any restrictions on data availability
- For clinical datasets or third party data, please ensure that the statement adheres to our policy

All mass spectrometry data have been deposited to the ProteomeXchange Consortium via the PRIDE partner repository with the dataset identifiers PXD036784 and PXD036945. Proteomic analysis used the Uniprot E. coli reference proteome (UP000000625).

# Research involving human participants, their data, or biological material

Policy information about studies with human participants or human data. See also policy information about sex, gender (identity/presentation), and sexual orientation and race, ethnicity and racism.

| | |
|---|---|
| Reporting on sex and gender | Not applicable |
| Reporting on race, ethnicity, or other socially relevant groupings | Not applicable |
| Population characteristics | Not applicable |
| Recruitment | Not applicable |
| Ethics oversight | Not applicable |

Note that full information on the approval of the study protocol must also be provided in the manuscript.

# Field-specific reporting

Please select the one below that is the best fit for your research. If you are not sure, read the appropriate sections before making your selection.

☒ Life sciences   ☐ Behavioural & social sciences   ☐ Ecological, evolutionary & environmental sciences

For a reference copy of the document with all sections, see nature.com/documents/nr-reporting-summary-flat.pdf

# Life sciences study design

All studies must disclose on these points even when the disclosure is negative.

| | |
|---|---|
| Sample size | No statistical methods were used to calculate sample sizes. Biochemical experiments were repeated at least three times, and in some cases using independent protein purifications. 2-4 replicates of HDX-MS data were collected to account for technical variability in pipetting, considering prior evidence in the literature supporting the high technical reproducibility of these experiments, and consistent with community guidelines for HDX MS (Masson, G. R. et al. Recommendations for performing, interpreting and reporting hydrogen deuterium exchange mass spectrometry (HDX-MS) experiments. Nat Methods 16, 595–602 (2019)). In some cases, replicate data were also acquired for independent purifications of the protein complexes to assess biological variability. For other experiments (enzyme assays, pelleting assays), the high reproducibility and large effect sizes indicate that the number of replicates was sufficient. |
| Data exclusions | No data were excluded. |
| Replication | All experiments were confirmed using replicate measurements. |
| Randomization | No randomization was performed, as samples were not grouped. |
| Blinding | No blinding was performed, as samples were not grouped. Moreover, MS data were analysed using standard pipelines which minimise the possibility of subjective interpretation. |

# Reporting for specific materials, systems and methods

We require information from authors about some types of materials, experimental systems and methods used in many studies. Here, indicate whether each material, system or method listed is relevant to your study. If you are not sure if a list item applies to your research, read the appropriate section before selecting a response.

## Materials & experimental systems

| n/a | Involved in the study |
|---|---|
| ☐ | ☒ Antibodies |
| ☒ | ☐ Eukaryotic cell lines |
| ☒ | ☐ Palaeontology and archaeology |
| ☒ | ☐ Animals and other organisms |
| ☒ | ☐ Clinical data |
| ☒ | ☐ Dual use research of concern |
| ☒ | ☐ Plants |

## Methods

| n/a | Involved in the study |
|---|---|
| ☒ | ☐ ChIP-seq |
| ☒ | ☐ Flow cytometry |
| ☒ | ☐ MRI-based neuroimaging |

## Antibodies

| | |
|---|---|
| Antibodies used | Anti Trigger factor - A01329, GenScript, polyclonal. Diluted 1:1000 |
| Validation | Eunyong Park and Tom A. Rapoport., et al. Bacterial Protein Translocation Requires Only One Copy Of The Secy Complex In Vivo. J Cell Biol. (2012-09)<br>Also validated in the current manuscript using purified Trigger factor (Extended Data Figure 6f). |

## Plants

| | |
|---|---|
| Seed stocks | *Report on the source of all seed stocks or other plant material used. If applicable, state the seed stock centre and catalogue number. If plant specimens were collected from the field, describe the collection location, date and sampling procedures.* |
| Novel plant genotypes | *Describe the methods by which all novel plant genotypes were produced. This includes those generated by transgenic approaches, gene editing, chemical/radiation-based mutagenesis and hybridization. For transgenic lines, describe the transformation method, the number of independent lines analyzed and the generation upon which experiments were performed. For gene-edited lines, describe the editor used, the endogenous sequence targeted for editing, the targeting guide RNA sequence (if applicable) and how the editor was applied.* |
| Authentication | *Describe any authentication procedures for each seed stock used or novel genotype generated. Describe any experiments used to assess the effect of a mutation and, where applicable, how potential secondary effects (e.g. second site T-DNA insertions, mosiacism, off-target gene editing) were examined.* |

