## [Peer Review File · Nature Structural & Molecular Biology]

Peer Review Information

Manuscript Title: Resolving chaperone-assisted protein folding on the ribosome at the peptide level

Corresponding author name(s): David Balchin, John Engen

Editorial Notes:

Transferred manuscripts This manuscript has been previously reviewed at another journal that is not operating a transparent peer review scheme. This document only contains reviewer comments, rebuttal and decision letters for versions considered at Nature Structural & Molecular Biology.

Reviewer Comments & Decisions:

Decision Letter, initial version:

Message: 19th Oct 2023

Dear Dr. Balchin,

Thank you again for submitting your manuscript "Resolving chaperone-assisted protein folding on the ribosome at the peptide level". I sincerely apologize for the unusual delay in responding, which resulted from the difficulty in obtaining suitable referee reports. Nevertheless, we now have comments (below) from the 2 reviewers who evaluated your paper, which includes the original Reviewer #2, and a new Reviewer #4, who has assessed the revised manuscript in reference to the concerns raised by the original Reviewers #1 and #3. In light of those reports, we remain interested in your study and would like to see your response to the comments of the referees, in the form of a revised manuscript.

You will see that while both reviewers appreciate the value of the work, Reviewer #2 still has concerns regarding addressing some remaining open mechanistic and interpretational questions, as well as clarifying some of the results, which should be addressed before we can make a final decision. Please be sure to address/respond to all concerns of the referees in full in a point-by-point response and highlight all changes in the revised manuscript text file. If you have comments that are intended for editors only, please

include those in a separate cover letter.

We expect to see your revised manuscript within 6 weeks. If you cannot send it within this time, please contact us to discuss an extension; we would still consider your revision, provided that no similar work has been accepted for publication at NSMB or published elsewhere.

Reporting Summary:

When submitting the revised version of your manuscript, please pay close attention to our [href="https://www.nature.com/nature-portfolio/editorial-policies/image-integrity">Digital Image Integrity Guidelines](https://www.nature.com/nature-portfolio/editorial-policies/image-integrity). and to the following points below:

Please note that all key data shown in the main figures as cropped gels or blots should be presented in uncropped form, with molecular weight markers. These data can be aggregated into a single supplementary figure item. While these data can be displayed in a relatively informal style, they must refer back to the relevant figures. These data should be submitted with the final revision, as source data, prior to acceptance, but you may want to start putting it together at this point.

SOURCE DATA: we request that authors provide, in tabular form, the data underlying the graphical representations used in figures. This is to further increase transparency in data reporting, as detailed in this editorial

(<http://www.nature.com/nsmb/journal/v22/n10/full/nsmb.3110.html>).
Spreadsheets can be submitted in excel format. Only one (1) file per figure is permitted; thus, for multi-paneled figures, the source data for each panel should be clearly labeled in the Excel file; alternately the data can be provided as multiple, clearly labeled sheets in an Excel file. When submitting files, the title field should indicate which figure the source data pertains to. We encourage our authors to provide source data at the revision stage, so that they are part of the peer-review process.

Data availability: this journal strongly supports public availability of data. All data used in accepted papers should be available via a public data repository, or alternatively, as Supplementary Information. If data can only be shared on request, please explain why in your Data Availability Statement, and also in the correspondence with your editor. Please note that for some data types, deposition in a public repository is mandatory - more information on our data deposition policies and available repositories can be found below: <https://www.nature.com/nature-research/editorial-policies/reporting-standards#availability-of-data>

We require deposition of coordinates (and, in the case of crystal structures, structure factors) into the Protein Data Bank with the designation of immediate release upon publication (HPUB). Electron microscopy-derived density maps and coordinate data must be deposited in EMDDB and released upon publication. Deposition and immediate release of NMR chemical shift assignments are highly encouraged. Deposition of deep sequencing and microarray data is mandatory, and the datasets must be released prior to or upon publication. To avoid delays in publication, dataset accession numbers must be supplied with the final accepted manuscript and appropriate release dates must be indicated at the galley proof stage.

Nature Structural & Molecular Biology is committed to improving transparency in authorship. As part of our efforts in this direction, we are now requesting that all authors identified as 'corresponding author' on published papers create and link their Open Researcher and Contributor Identifier (ORCID) with their account on the Manuscript Tracking System (MTS), prior to acceptance. This applies to primary research papers only. ORCID helps the scientific community achieve unambiguous attribution of all scholarly contributions. You can create and link your ORCID from the home page of the MTS by clicking on 'Modify my Springer Nature account'. For more information please visit please visit www.springernature.com/orcid.

[Redacted]

We look forward to seeing the revised manuscript and thank you for the opportunity to

review your work.

Sincerely,
Sara

Sara Osman, Ph.D.
Associate Editor
Nature Structural & Molecular Biology

Reviewers' Comments:

Reviewer #2:
Remarks to the Author:
Attached

Reviewer #4:
Remarks to the Author:
NSMB-A48022-T

This is an exciting paper that presents a comprehensive study of DHFR folding on the ribosome. By stopping the protein synthesis at four different locations, the authors are able to sample the foldedness of DHFR as it emerges from the ribosome. An important aspect of this study is the characterization of the role of Trigger factor. The collaborative group of authors encompasses experts in ribosome preparation with stalled nascent chains as well as HDX-MS, the primary approach to characterizing the equilibrium state of the folding intermediates. This work represents a different approach from the usual kinetic trapping that is used for HDX characterization of folding intermediates in which pulse-chase is used to sample the folded states along the kinetic pathway. Here, the authors set-up the folding intermediates by stalling the ribosome and then sample their equilibrium structures. The approach allows them to gain a lot of information about the intermediate folded states of DHFR as it emerges from the ribosome and they do a good job of putting their results in context with other kinetic approaches. They performed all the important control experiments including HDX-MS of the native state DHFR and compare to previous studies on in vitro folding of DHFR from urea denaturation.

I have read the previous reviews and I think the authors did an excellent job of addressing all the reviewer's comments (many of which I agree with). The remaining issues, such as the slight decrease in deuteration over longer time points and the discrepancies in the Max D deuteration are not uncommon in HDX-MS experiments and do not affect the conclusions of the study.

The results demonstrate several important findings. 1) The folding pathway is different for DHFR emerging from the ribosome as compared to the full-length protein folding from urea. 2) The N-terminal 37 residues of DHFR is more "protected" than would be expected for an unfolded state indicating that it is interacting with the ribosome exit tunnel. 3) By comparing various constructs of the C-terminal domain confined in the exit tunnel or linked to a flexible linker, they show that DHFR instability is induced by unstructured C-termini which is rescued by confinement in the ribosome exit tunnel. 4) DHFR is more active when still attached to the ribosome (a LOT of additional experiments were done to demonstrate statistical significance of this result albeit the difference is only 2-fold).

I think the most exciting part of this work is the characterization of the interactions of

Trigger Factor which co-purified with several of the ribosome-nascent chain complexes. The hydrophobic parts of TF apparently weakly dimerize in solution, but when bound to the ribosome they interact with hydrophobic parts of the NC as shown by strong HDX-MS protection in the TF-RNC complex at both hydrophobic and hydrophilic regions of TF. Finally, the authors were able to see protection of the ribosomal proteins lining the exit tunnel demonstrating conclusively that the NC interacts with the exit tunnel surfaces.

Reviewer 1 Feedback:

HDXMS is used here to probe DHFR during synthesis on the ribosome. The value of this manuscript is that it paves a way for a mass spec/proteomics method that can be used to understand structure and dynamics of translation. The authors consider progressive synthesis of DHFR and are able to provide information on the nascent polypeptide, the presence/absence of trigger factor, ribosomal proteins. The hyperactivity observations of DHFR on the ribosome (e.g., in the FL+58 RNC) are nice (albeit not really explained) added to which the trigger factor and ribosomal protein exchange analyses are interesting and reveal strengths of the method. This is highly rigorous work and the authors have developed a very elegant strategy that can look at many system components in a dynamic system simultaneously. Unfortunately, while the ability to look at the trigger factor and the ribosomal proteins is neat, the main problem that the cot pathway of DHFR is actually *not* at all clearly described.

The N-terminal DLD region emerges and is disordered and then just 4 peptides (among the very many possible) are used to suggest native-like ABD compaction to form a putative cot intermediate not seen in isolation. At nascent chain lengths where this protection is seen the trigger factor is recruited in the cell but similar protection levels are also seen in TF deletion strains. While appealing, this evidence alone for the natively-structured ABD intermediate appears to be insufficient. However, this is also the case for both the prior NMR and FRET work on nascent chains as this is technically highly challenging work. Nascent chain interactions with the ribosome could replace those of the TF, as could a simple non-native or a partial compaction of a disordered state of this (hydrophobic?) region, or any combination of these, and could be the basis of the observed ABD domain protection. The very few peptides that simply show similar uptake of deuterium as the indicator of this intermediate (Fig 3b) are incomplete evidence as they report only on protection with no other structural signature. The 106-RNC and the 126-RNC are the two RNCs on which many conclusions rest. They allow investigation of the ABD protection and the extent of protection looks very similar in both peptides. Is the deduction of native-like folding of this domain even feasible when, in at least one of these (1-106-RNC), the entire ABD subdomain is unavailable through its tunnel occlusion? This seems unlikely, especially when the idea is that the N-terminal DLD is unfolded and not involved.

So, while Fig 2 on the full-length emerged constructs have many data points, show similarities throughout the sequence and come together to persuade on native-like structure, especially together with activity data, those of the emerging lengths on which the intermediate arguments rest are weaker. The result is that 'a complete co-translational folding pathway' is actually not defined but is instead hinted at.

When compared, the cot folding pathway does show apparent differences to the in vitro pathway referenced (Ref Matthews, 1995) of refolding of DHFR from denaturant, where at least some of the DLD stranding network forms early in the refolding experiments. The structural signatures of prior structural in vitro work are detailed but also involve only full length DHFR rather than truncations that correspond to the vectorial case on the ribosome. As an aside, while the isolated truncations used in this study appear insoluble even

in the absence of a GFP fusion, apart from in one case, they do remain soluble on the ribosome. This would seem to me to be a solution for their investigation i.e., isolated in the presence of the ribosome. This may allow the investigation of the impact of the ribosome in stabilizing the ABD domain.

This could be a misreading but the construct lengths used appear to not investigate how the C-terminal DLD region might perturb the equilibria: 1-126 RNC has this region in the tunnel and the subsequent RNC length shown is FL+58 where there is native like folding across the board. RNC lengths in between these would allow some of the details to be fleshed out (cf the 106-RNC mention above where ABD protection occurs even with partial tunnel emergence). An added peptide can result in folding but, just preceding this, complete folding are there any hints of interactions of this region? Within the N-terminal DLD, is this segment expanded or a compact disordered state? Can the method distinguish these? If not can other methods (e.g., NMR, FRET) be used to provide more contemporary structural detail? Has any modelling been undertaken for a dynamic system?

How clear are the tunnel/protein availability boundaries? This is generally not a concern if the sequence in the tunnel is known to be either in an extended conformation from cryoE-M or even better constant so that only

1

differences are being described, but this is not the case here. Have surface exposure experiments been undertaken? How can the differences between FL+58 and FL+38 be explained? Do the HDX data for FL+28, FL+38 and FL+58 offer further structural insights e.g., about the conformational properties of the C-terminal DLD region. This would seem to be the case and the FL+38 data in Extended Fig 5 is interesting with the DLD1 showing greater protection than DLD2. These could be further discussed.

“Furthermore, we observed extensive deprotection (relative to DHFR+50stop) relative to native FL DHFR, consistent with increased structural dynamics (Fig.3e and Extended Data Fig. 4c-e). Deprotection was strongest in peptides covering β 1-3, α 1 and α 2, indicating that neither the DLD nor ABD is as stably folded as in FL+50stop.” There are certainly some differences but these are rather small (rather than extensive) for any real inferences surely. The dotted lines can be misleading as there is no data on many peptides such that this should not be suggested at.

Overall,

- (1) for a high-end technique, the overall description of the cot folding pathway is too qualitative, and overreaches at times. The data are not robust enough to independently report on cot phenomena
- (2) the biological conclusions are somewhat underwhelming and do not add significantly to the knowledge of cot;
- (3) I feel that the quantitative potential of this powerful method has not been fully demonstrated here (e.g., cf a related mass spec paper deriving stabilities using methionine oxidation <https://pubmed.ncbi.nlm.nih.gov/37552756/>, also studied DHFR).

6

Author Rebuttal to Initial comments

Reviewer #2

HDXMS is used here to probe DHFR during synthesis on the ribosome. The value of this manuscript is that it paves a way for a mass spec/proteomics method that can be used to understand structure and dynamics of translation. The authors consider progressive synthesis of DHFR and are able to provide information on the nascent polypeptide, the presence/absence of trigger factor, ribosomal proteins. The hyperactivity observations of DHFR on the ribosome (e.g., in the FL+58 RNC) are nice (albeit not really explained) added to which the trigger factor and ribosomal protein exchange analyses are interesting and reveal strengths of the method. This is highly rigorous work and the authors have developed a very elegant strategy that can look at many system components in a dynamic system simultaneously. Unfortunately, while the ability to look at the trigger factor and the ribosomal proteins is neat, the main problem that the cot pathway of DHFR is actually not at all clearly described.

We appreciate the recognition of the unique strengths of our experimental approach. We think that simultaneous analysis of the NC and bound chaperones is an important step towards a more authentic model of cotranslational folding.

Below, the reviewer introduces a series of new critiques of the revised manuscript, centred around the argument that the folding pathway of DHFR is poorly described by our HDX-MS data. We have endeavoured to address these in detail, with a particular focus on explaining how the HDX-MS data are interpreted.

The N-terminal DLD region emerges and is disordered and then just 4 peptides (among the very many possible) are used to suggest native-like ABD compaction to form a putative cot intermediate not seen in isolation. At nascent chain lengths where this protection is seen the trigger factor is recruited in the cell but similar protection levels are also seen in TF deletion strains. While appealing, this evidence alone for the natively-structured ABD intermediate appears to be insufficient. However, this is also the case for both the prior NMR and FRET work on nascent chains as this is technically highly challenging work. Nascent chain interactions with the ribosome could replace those of the TF, as could a simple non-native or a partial compaction of a disordered state of this (hydrophobic?) region, or any combination of these, and could be the basis of the observed ABD domain protection. The very few peptides that simply show similar uptake of deuterium as the indicator of this intermediate (Fig 3b) are incomplete evidence as they report only on protection with no other structural signature.

The reviewer argues that protection of the ABD is insufficient evidence to claim that this subdomain is folded in the RNCs. This argument reflects a misunderstanding about what exactly is measured by HDX-MS, which is explained in line 152 of the manuscript.

We agree that simple “protection” would indeed not definitively indicate folding. However, the experiment does not measure “protection”, but rather quantifies deuterium uptake. Thus, folding to the native state can be assessed by quantitatively comparing deuterium uptake (in Da, with a precision better than 0.25 mass units) in the NC to that measured for the native controls (FL DHFR and FL+50^{RNC}). As pointed out by Reviewer #4, this is analogous to a pulsed-label HDX-MS experiment which is commonly used to follow folding over time. In fact, using stalled RNCs allows us to make an even more rigorous comparison with the native state, since we can sample different durations of deuterium exposure (in this case 10, 100, 1000 sec).

Figure 3A nicely illustrates this point. Although 1-37^{RNC} is protected relative to e.g. 1-64^{RNC}, the level of deuteration in the NC (at three independent time points) is very easily distinguishable from the level of deuteration in the native state controls. To mimic native folding, interactions (e.g. with the ribosome) or “partial compaction” would have to precisely mimic the number of satisfied backbone hydrogen bonds found in native DHFR. Note that amide hydrogen bonding (and to a lesser extent, solvent accessibility) is the primary determinant of HDX kinetics.

In our experiments on 1-106^{RNC} and 1-126^{RNC}, the ABD is covered by 4 unique peptides. Importantly, this includes the 4 β -strands comprising the core of this subdomain. For each peptide, deuterium uptake is identical across three time points between the NC and native state controls, allowing us to conclude that the regions covered by these peptides are native-like in their conformation.

The 106-RNC and the 126-RNC are the two RNCs on which many conclusions rest. They allow investigation of the ABD protection and the extent of protection looks very similar in both peptides. Is the deduction of native-like folding of this domain even feasible when, in at least one of these (1-106-RNC), the entire ABD subdomain is unavailable through its tunnel occlusion? This seems unlikely, especially when the idea is that the N-terminal DLD is unfolded and not involved.

An important advantage of HDX-MS is that folding can be probed locally, and it is true that our data show that part of the ABD folds even before the complete subdomain emerges from the exit tunnel.

As shown in Figure 1A, the ABD consists of four β -strands (β 2- β 5) and three α -helices (α 2- α 4). As shown in Figure 3C, 1-106^{RNC} is expected to expose ~84 residues, comprising a continuous 3-stranded β -sheet (β 2- β 4) with two flanking helices (α 2- α 3). Thus, from a structural perspective, folding of this large fragment of the ABD is not surprising. Many autonomously folding domains are much smaller than this - see <https://doi.org/10.1038/s41586-023-06328-6> for >400 examples of folded domains 40–72 amino acids in length.

Indeed, it would be more surprising if folding of this 84-residue fragment was dependent on exposure of the complete ABD, since the missing β 5 is very short – just 2 residues.

So, while Fig 2 on the full-length emerged constructs have many data points, show similarities throughout the sequence and come together to persuade on native-like structure, especially together with activity data, those of the emerging lengths on which the intermediate arguments rest are weaker. The result is that ‘a complete co-translational folding pathway’ is actually not defined but is instead hinted at.

In addition to the arguments above, we would like to make it clear that the absolute number of peptides is not a critical factor in our interpretation. Each peptide is independent, and the deuteration of each robustly reports on the local amide hydrogen environment for that region of DHFR.

When compared, the cot folding pathway does show apparent differences to the in vitro pathway referenced (Ref Matthews, 1995) of refolding of DHFR from denaturant, where at least some of the DLD stranding network forms early in the refolding experiments. The structural signatures of prior structural in vitro work are detailed but also involve only full length DHFR rather than truncations that correspond to the vectorial case on the ribosome.

We agree, and of course this is a key point that we make explicit in the manuscript: vectorial synthesis changes the folding pathway.

As an aside, while the isolated truncations used in this study appear insoluble even in the absence of a GFP fusion, apart from in one case, they do remain soluble on the ribosome. This would seem to me to be a solution for their investigation i.e., isolated in the presence of the ribosome. This may allow the investigation of the impact of the ribosome in stabilizing the ABD domain.

This is an interesting idea, but rather different to the established model for ribosome-induced solubility (ref. 71), where the ribosome acts like a solubility tag in a fusion protein. We are not aware of any evidence suggesting that simple crowding by ribosomes suffices to solubilise aggregation-prone proteins.

This could be a misreading but the construct lengths used appear to not investigate how the C-terminal DLD region might perturb the equilibria: 1-126 RNC has this region in the tunnel and the subsequent RNC length shown is FL+58 where there is native like folding across the board. RNC lengths in between these would allow some of the details to be fleshed out (cf the 106-RNC mention above where ABD protection occurs even with partial tunnel emergence).

The reviewer is correct – our RNCs do not sample a scenario where only part of the C-terminal half of the DLD is out of the ribosome. Given that the C-terminal sequence forms a strand that inserts in the centre of the DLD, it is unlikely that this subdomain would fold partially during synthesis. We cannot rule this out, however, and therefore do not comment on this possibility in the manuscript.

In general, although it would always be desirable to study more RNCs, we have to balance this against the considerable effort of doing so, taking into account the potential for new biological insight.

An added peptide can result in folding but, just preceding this, complete folding are there any hints of interactions of this region? Within the N-terminal DLD, is this segment expanded or a compact disordered state? Can the method distinguish these? If not can other methods (e.g., NMR, FRET) be used to provide more contemporary structural detail? Has any modelling been undertaken for a dynamic system?

There is perhaps a very slight (~0.4 Da at 10 sec exposure time) decrease in uptake of the N-terminal region when the NC elongates from 106 to 126 residues (Fig. 3A and Table S1). This is definitely not native folding (~4 Da decrease in deuteration at 10 sec) but could reflect subtle compaction of the NC. Although HDX is highly sensitive to folding events (in particular folding of secondary structure), it is not well suited to probing the degree of compaction of disordered ensembles. Since our focus was on folding transitions, we chose not to pursue this question further.

How clear are the tunnel/protein availability boundaries? This is generally not a concern if the sequence in the tunnel is known to be either in an extended conformation from cryoE-M or even better constant so that only 2 differences are being described, but this is not the case here. Have surface exposure experiments been undertaken?

Unfortunately, there is no definitive way to calculate this precisely. Techniques with the necessary structural resolution (cryo-EM/x-ray crystallography) do not account for structural heterogeneity in the NC. The tunnel boundaries we describe are estimates based on commonly used values from prior literature (i.e. 30aa required to span the tunnel), and we

were careful to describe them as such in the manuscript. It is certainly possible that the real tunnel boundaries differ slightly from the estimates. However, we think that our interpretation is insensitive to the true tunnel boundary, for two reasons. 1) Our conclusions are always based on more than one chain length. E.g. 1-37/64 reporting on the DLD and 1-106/126 reporting on the ABD. 2) Since the HDX data are locally resolved, we can comment on folding regardless of how close the specific peptide is to the ribosome. Thus, our conclusions do not depend on knowing exactly how much of the NC is exposed outside the ribosome at each stalling position.

How can the differences between FL+58 and FL+38 be explained?

In FL+38 we observe protection/deprotection mostly at peripheral loops in DHFR. As we mention in the manuscript (line 294), DHFR is expected to be much closer to the ribosome surface in FL+38 compared to FL+58. The difference in DHFR conformation might arise from interactions with the ribosome surface, as previously described for other NCs (see the Discussion section).

Do the HDX data for FL+28, FL+38 and FL+58 offer further structural insights e.g., about the conformational properties of the C-terminal DLD region. This would seem to be the case and the FL+38 data in Extended Fig 5 is interesting with the DLD1 showing greater protection than DLD2. These could be further discussed.

In FL+38, we observe that part of DLD1 is protected relative to FL DHFR, while part of DLD2 is deprotected. We can only speculate, but one possibility is that, by binding weakly to DLD1, the ribosome destabilizes the interaction between DLD2 and DLD1.

In general, we have chosen not to emphasize this aspect as it relates to the effect of the ribosome on the conformational stability of a tethered folded domain, rather than the cotranslational domain folding pathway which is our focus.

“Furthermore, we observed extensive deprotection (relative to DHFR+50stop) relative to native FL DHFR, consistent with increased structural dynamics (Fig.3e and Extended Data Fig. 4c-e). Deprotection was strongest in peptides covering β 1-3, α 1 and α 2, indicating that neither the DLD nor ABD is as stably folded as in FL+50stop.” There are certainly some differences but these are rather small (rather than extensive) for any real inferences surely. The dotted lines can be misleading as there is no data on many peptides such that this should not be suggested at.

“Extensive” meaning “covering a large area” was used to communicate that a large portion of the protein was deprotected, rather than to convey the degree of deprotection. The extent of the deprotection is clear in Figure 3E – most of the protein is deprotected to some degree.

The differences in uptake range from 0.5 to 1.5 D. This is well outside the precision of the mass measurements, and considered in the HDX-MS field to be substantial differences. Of course, these data do not suggest that FL+50stop is unfolded, but it is clearly conformationally destabilized relative to FL DHFR.

We are not sure why the reviewer thinks that “there are no data on many peptides”. Each data point in Extended data Fig. 4c-e represents a peptide, which is listed on the x-axis. For these two proteins (FL+50stop and FL DHFR) we measured deuterium uptake for 97 peptides covering 97% of the DHFR sequence (Data S1).

Overall,

(1) for a high-end technique, the overall description of the cot folding pathway is too qualitative, and overreaches at times. The data are not robust enough to independently report on cot phenomena

We respectfully disagree with the reviewer. We have collected and reported quantitative HDX-MS data, then additionally described the key findings qualitatively. The fact that the folding pathway is clearly summarized as a stepwise sequence does not entail that the conclusions are not supported by quantitative data. For example, Fig 3 shows a detailed quantitative comparison of deuterium uptake for peptides in each RNC, followed by an assessment of the meaning of uptake differences which is displayed on the protein structure.

We do not see any evidence in the review to justify the claim that “The data are not robust enough to independently report on cot phenomena”.

(2) the biological conclusions are somewhat underwhelming and do not add significantly to the knowledge of cot;

We have addressed this point in detail in the previous rebuttal letter, including extensively referring to the literature.

(3) I feel that the quantitative potential of this powerful method has not been fully demonstrated here (e.g., cf a related mass spec paper deriving stabilities using methionine oxidation <https://pubmed.ncbi.nlm.nih.gov/37552756/>, also studied DHFR).

In the paper referred to by the reviewer, methionine oxidation is used to probe NC tertiary structure upon chemical denaturation, showing differences in thermodynamic stability between released and ribosome-tethered full-length proteins. While this is an interesting approach, it probes distinct properties of protein conformation compared to HDX-MS, and addresses a quite different question to our study. As described in the introduction and discussion sections of our manuscript, our goal was to characterise the sequence of folding events that occur during domain synthesis, and the role of Trigger factor.

Reviewer #4:

This is an exciting paper that presents a comprehensive study of DHFR folding on the ribosome. By stopping the protein synthesis at four different locations, the authors are able to sample the foldedness of DHFR as it emerges from the ribosome. An important aspect of this study is the characterization of the role of Trigger factor. The collaborative group of authors encompasses experts in ribosome preparation with stalled nascent chains as well as HDX-MS, the primary approach to characterizing the equilibrium state of the folding intermediates. This work represents a different approach from the usual kinetic trapping that is used for HDX characterization of folding intermediates in which pulse-chase is used to sample the folded states along the kinetic pathway. Here, the authors set-up the folding intermediates by stalling the ribosome and then sample their equilibrium structures. The approach allows them to gain a lot of information about the intermediate folded states of DHFR as it emerges from the ribosome and they do a good job of putting their results in context with other kinetic approaches. They performed all the important control experiments including HDX-MS of the native state DHFR and compare to previous studies on in vitro folding of DHFR from urea denaturation.

I have read the previous reviews and I think the authors did an excellent job of addressing all the reviewer's comments (many of which I agree with). The remaining issues, such as the slight decrease in deuteration over longer time points and the discrepancies in the Max D

deuteration are not uncommon in HDX-MS experiments and do not affect the conclusions of the study.

The results demonstrate several important findings. 1) The folding pathway is different for DHFR emerging from the ribosome as compared to the full-length protein folding from urea. 2) The N-terminal 37 residues of DHFR is more “protected” than would be expected for an unfolded state indicating that it is interacting with the ribosome exit tunnel. 3) By comparing various constructs of the C-terminal domain confined in the exit tunnel or linked to a flexible linker, they show that DHFR instability is induced by unstructured C-termini which is rescued by confinement in the ribosome exit tunnel. 4) DHFR is more active when still attached to the ribosome (a LOT of additional experiments were done to demonstrate statistical significance of this result albeit the difference is only 2-fold).

I think the most exciting part of this work is the characterization of the interactions of Trigger Factor which co-purified with several of the ribosome-nascent chain complexes. The hydrophobic parts of TF apparently weakly dimerize in solution, but when bound to the ribosome they interact with hydrophobic parts of the NC as shown by strong HDX-MS protection in the TF-RNC complex at both hydrophobic and hydrophilic regions of TF. Finally, the authors were able to see protection of the ribosomal proteins lining the exit tunnel demonstrating conclusively that the NC interacts with the exit tunnel surfaces.

We thank the reviewer for their positive assessment of our work, and for taking the time to carefully evaluate the previous reviews/rebuttal.

Decision Letter, first revision:

Message: Our ref: NSMB-A48022A

22nd Feb 2024

Dear Dr. Balchin,

Thank you for submitting your revised manuscript "Resolving chaperone-assisted protein folding on the ribosome at the peptide level" (NSMB-A48022A). I sincerely apologize for the delay in responding. The editorial team has decided to step in and assess the manuscript in-house, based on which we have decided that we'll be happy in principle to publish it in Nature Structural & Molecular Biology, conditional on including discussion on the remaining referee concerns. We would also like to encourage you to opt in to our Transparent Peer Review option, as conveying the interpretational arguments from both sides will be very informative for the readers, and pending also minor revisions to comply with our editorial and formatting guidelines.

We are now performing detailed checks on your paper and will send you a checklist detailing our editorial and formatting requirements in the next few weeks. Please do not upload the final materials and make any revisions until you receive this additional information from us.

To facilitate our work at this stage, it is important that we have a copy of the main text as a word file. If you could please send along a word version of this file as soon as possible, we would greatly appreciate it; please make sure to copy the NSMB account (cc'ed above).

Sincerely,
Sara

Sara Osman, Ph.D.
Associate Editor
Nature Structural & Molecular Biology

Author Rebuttal, first revision:

Reviewer #2

HDXMS is used here to probe DHFR during synthesis on the ribosome. The value of this manuscript is that it paves a way for a mass spec/proteomics method that can be used to understand structure and dynamics of translation. The authors consider progressive synthesis of DHFR and are able to provide information on the nascent polypeptide, the presence/absence of trigger factor, ribosomal proteins. The hyperactivity observations of DHFR on the ribosome (e.g., in the FL+58 RNC) are nice (albeit not really explained) added to which the trigger factor and ribosomal protein exchange analyses are interesting and reveal strengths of the method. This is highly rigorous work and the authors have developed a very elegant strategy that can look at many system components in a dynamic system simultaneously. Unfortunately, while the ability to look at the trigger factor and the ribosomal proteins is neat, the main problem that the cot pathway of DHFR is actually not at all clearly described.

We appreciate the recognition of the unique strengths of our experimental approach. We think that simultaneous analysis of the NC and bound chaperones is an important step towards a more authentic model of cotranslational folding.

Below, the reviewer introduces a series of new critiques of the revised manuscript, centred around the argument that the folding pathway of DHFR is poorly described by our HDX-MS data. We have endeavoured to address these in detail, with a particular focus on explaining how the HDX-MS data are interpreted.

The N-terminal DLD region emerges and is disordered and then just 4 peptides (among the very many possible) are used to suggest native-like ABD compaction to form a putative cot intermediate not seen in isolation. At nascent chain lengths where this protection is seen the trigger factor is recruited in the cell but similar protection levels are also seen in TF deletion strains. While appealing, this evidence alone for the natively-structured ABD intermediate appears to be insufficient. However, this is also the case for both the prior NMR and FRET work on nascent chains as this is technically highly challenging work. Nascent chain interactions with the ribosome could replace those of the TF, as could a simple non-native or a partial compaction of a disordered state of this (hydrophobic?) region, or any combination of these, and could be the basis of the observed ABD domain protection. The very few peptides that simply show similar uptake of deuterium as the indicator of this intermediate (Fig 3b) are incomplete evidence as they report only on protection with no other structural signature.

The reviewer argues that protection of the ABD is insufficient evidence to claim that this subdomain is folded in the RNCs. This argument reflects a misunderstanding about what exactly is measured by HDX-MS, which is explained in line 152 of the manuscript.

We agree that simple “protection” would indeed not definitively indicate folding. However, the experiment does not measure “protection”, but rather quantifies deuterium uptake. Thus, folding to the native state can be assessed by quantitatively comparing deuterium uptake (in Da, with a precision better than 0.25 mass units) in the NC to that measured for the native controls (FL DHFR and FL+50^{RNC}). As pointed out by Reviewer #4, this is analogous to a pulsed-label HDX-MS experiment which is commonly used to follow folding over time. In fact, using stalled RNCs allows us to make an even more rigorous comparison with the native state, since we can sample different durations of deuterium exposure (in this case 10, 100, 1000 sec).

Figure 3A nicely illustrates this point. Although 1-37^{RNC} is protected relative to e.g. 1-64^{RNC}, the level of deuteration in the NC (at three independent time points) is very easily distinguishable from the level of deuteration in the native state controls. To mimic native folding, interactions (e.g. with the ribosome) or “partial compaction” would have to precisely mimic the number of satisfied backbone hydrogen bonds found in native DHFR. Note that amide hydrogen bonding (and to a lesser extent, solvent accessibility) is the primary determinant of HDX kinetics.

In our experiments on 1-106^{RNC} and 1-126^{RNC}, the ABD is covered by 4 unique peptides. Importantly, this includes the 4 β -strands comprising the core of this subdomain. For each peptide, deuterium uptake is identical across three time points between the NC and native state controls, allowing us to conclude that the regions covered by these peptides are native-like in their conformation.

The 106-RNC and the 126-RNC are the two RNCs on which many conclusions rest. They allow investigation of the ABD protection and the extent of protection looks very similar in both peptides. Is the deduction of native-like folding of this domain even feasible when, in at least one of these (1-106-RNC), the entire ABD subdomain is unavailable through its tunnel occlusion? This seems unlikely, especially when the idea is that the N-terminal DLD is unfolded and not involved.

An important advantage of HDX-MS is that folding can be probed locally, and it is true that our data show that part of the ABD folds even before the complete subdomain emerges from the exit tunnel.

As shown in Figure 1A, the ABD consists of four β -strands (β 2- β 5) and three α -helices (α 2- α 4). As shown in Figure 3C, 1-106^{RNC} is expected to expose ~84 residues, comprising a continuous 3-stranded β -sheet (β 2- β 4) with two flanking helices (α 2- α 3). Thus, from a structural perspective, folding of this large fragment of the ABD is not surprising. Many autonomously folding domains are much smaller than this - see <https://doi.org/10.1038/s41586-023-06328-6> for >400 examples of folded domains 40–72 amino acids in length.

Indeed, it would be more surprising if folding of this 84-residue fragment was dependent on exposure of the complete ABD, since the missing β 5 is very short – just 2 residues.

So, while Fig 2 on the full-length emerged constructs have many data points, show similarities throughout the sequence and come together to persuade on native-like structure, especially together with activity data, those of the emerging lengths on which the intermediate arguments rest are weaker. The result is that ‘a complete co-translational folding pathway’ is actually not defined but is instead hinted at.

In addition to the arguments above, we would like to make it clear that the absolute number of peptides is not a critical factor in our interpretation. Each peptide is independent, and the deuteration of each robustly reports on the local amide hydrogen environment for that region of DHFR.

When compared, the cot folding pathway does show apparent differences to the in vitro pathway referenced (Ref Matthews, 1995) of refolding of DHFR from denaturant, where at least some of the DLD stranding network forms early in the refolding experiments. The structural signatures of prior structural in vitro work are detailed but also involve only full length DHFR rather than truncations that correspond to the vectorial case on the ribosome.

We agree, and of course this is a key point that we make explicit in the manuscript: vectorial synthesis changes the folding pathway.

As an aside, while the isolated truncations used in this study appear insoluble even in the absence of a GFP fusion, apart from in one case, they do remain soluble on the ribosome. This would seem to me to be a solution for their investigation i.e., isolated in the presence of the ribosome. This may allow the investigation of the impact of the ribosome in stabilizing the ABD domain.

This is an interesting idea, but rather different to the established model for ribosome-induced solubility (ref. 71), where the ribosome acts like a solubility tag in a fusion protein. We are not aware of any evidence suggesting that simple crowding by ribosomes suffices to solubilise aggregation-prone proteins.

This could be a misreading but the construct lengths used appear to not investigate how the C-terminal DLD region might perturb the equilibria: 1-126 RNC has this region in the tunnel and the subsequent RNC length shown is FL+58 where there is native like folding across the board. RNC lengths in between these would allow some of the details to be fleshed out (cf the 106-RNC mention above where ABD protection occurs even with partial tunnel emergence).

The reviewer is correct – our RNCs do not sample a scenario where only part of the C-terminal half of the DLD is out of the ribosome. Given that the C-terminal sequence forms a strand that inserts in the centre of the DLD, it is unlikely that this subdomain would fold partially during synthesis. We cannot rule this out, however, and therefore do not comment on this possibility in the manuscript.

In general, although it would always be desirable to study more RNCs, we have to balance this against the considerable effort of doing so, taking into account the potential for new biological insight.

An added peptide can result in folding but, just preceding this, complete folding are there any hints of interactions of this region? Within the N-terminal DLD, is this segment expanded or a compact disordered state? Can the method distinguish these? If not can other methods (e.g., NMR, FRET) be used to provide more contemporary structural detail? Has any modelling been undertaken for a dynamic system?

There is perhaps a very slight (~0.4 Da at 10 sec exposure time) decrease in uptake of the N-terminal region when the NC elongates from 106 to 126 residues (Fig. 3A and Table S1). This is definitely not native folding (~4 Da decrease in deuteration at 10 sec) but could reflect subtle compaction of the NC. Although HDX is highly sensitive to folding events (in particular folding of secondary structure), it is not well suited to probing the degree of compaction of disordered ensembles. Since our focus was on folding transitions, we chose not to pursue this question further.

How clear are the tunnel/protein availability boundaries? This is generally not a concern if the sequence in the tunnel is known to be either in an extended conformation from cryoE-M or even better constant so that only 2 differences are being described, but this is not the case here. Have surface exposure experiments been undertaken?

Unfortunately, there is no definitive way to calculate this precisely. Techniques with the necessary structural resolution (cryo-EM/x-ray crystallography) do not account for structural heterogeneity in the NC. The tunnel boundaries we describe are estimates based on commonly used values from prior literature (i.e. 30aa required to span the tunnel), and we

were careful to describe them as such in the manuscript. It is certainly possible that the real tunnel boundaries differ slightly from the estimates. However, we think that our interpretation is insensitive to the true tunnel boundary, for two reasons. 1) Our conclusions are always based on more than one chain length. E.g. 1-37/64 reporting on the DLD and 1-106/126 reporting on the ABD. 2) Since the HDX data are locally resolved, we can comment on folding regardless of how close the specific peptide is to the ribosome. Thus, our conclusions do not depend on knowing exactly how much of the NC is exposed outside the ribosome at each stalling position.

How can the differences between FL+58 and FL+38 be explained?

In FL+38 we observe protection/deprotection mostly at peripheral loops in DHFR. As we mention in the manuscript (line 294), DHFR is expected to be much closer to the ribosome surface in FL+38 compared to FL+58. The difference in DHFR conformation might arise from interactions with the ribosome surface, as previously described for other NCs (see the Discussion section).

Do the HDX data for FL+28, FL+38 and FL+58 offer further structural insights e.g., about the conformational properties of the C-terminal DLD region. This would seem to be the case and the FL+38 data in Extended Fig 5 is interesting with the DLD1 showing greater protection than DLD2. These could be further discussed.

In FL+38, we observe that part of DLD1 is protected relative to FL DHFR, while part of DLD2 is deprotected. We can only speculate, but one possibility is that, by binding weakly to DLD1, the ribosome destabilizes the interaction between DLD2 and DLD1.

In general, we have chosen not to emphasize this aspect as it relates to the effect of the ribosome on the conformational stability of a tethered folded domain, rather than the cotranslational domain folding pathway which is our focus.

“Furthermore, we observed extensive deprotection (relative to DHFR+50stop) relative to native FL DHFR, consistent with increased structural dynamics (Fig.3e and Extended Data Fig. 4c-e). Deprotection was strongest in peptides covering β 1-3, α 1 and α 2, indicating that neither the DLD nor ABD is as stably folded as in FL+50stop.” There are certainly some differences but these are rather small (rather than extensive) for any real inferences surely. The dotted lines can be misleading as there is no data on many peptides such that this should not be suggested at.

“Extensive” meaning “covering a large area” was used to communicate that a large portion of the protein was deprotected, rather than to convey the degree of deprotection. The extent of the deprotection is clear in Figure 3E – most of the protein is deprotected to some degree.

The differences in uptake range from 0.5 to 1.5 D. This is well outside the precision of the mass measurements, and considered in the HDX-MS field to be substantial differences. Of course, these data do not suggest that FL+50stop is unfolded, but it is clearly conformationally destabilized relative to FL DHFR.

We are not sure why the reviewer thinks that “there are no data on many peptides”. Each data point in Extended data Fig. 4c-e represents a peptide, which is listed on the x-axis. For these two proteins (FL+50stop and FL DHFR) we measured deuterium uptake for 97 peptides covering 97% of the DHFR sequence (Data S1).

Overall,

(1) for a high-end technique, the overall description of the cot folding pathway is too qualitative, and overreaches at times. The data are not robust enough to independently report on cot phenomena

We respectfully disagree with the reviewer. We have collected and reported quantitative HDX-MS data, then additionally described the key findings qualitatively. The fact that the folding pathway is clearly summarized as a stepwise sequence does not entail that the conclusions are not supported by quantitative data. For example, Fig 3 shows a detailed quantitative comparison of deuterium uptake for peptides in each RNC, followed by an assessment of the meaning of uptake differences which is displayed on the protein structure.

We do not see any evidence in the review to justify the claim that “The data are not robust enough to independently report on cot phenomena”.

(2) the biological conclusions are somewhat underwhelming and do not add significantly to the knowledge of cot;

We have addressed this point in detail in the previous rebuttal letter, including extensively referring to the literature.

(3) I feel that the quantitative potential of this powerful method has not been fully demonstrated here (e.g., cf a related mass spec paper deriving stabilities using methionine oxidation <https://pubmed.ncbi.nlm.nih.gov/37552756/>, also studied DHFR).

In the paper referred to by the reviewer, methionine oxidation is used to probe NC tertiary structure upon chemical denaturation, showing differences in thermodynamic stability between released and ribosome-tethered full-length proteins. While this is an interesting approach, it probes distinct properties of protein conformation compared to HDX-MS, and addresses a quite different question to our study. As described in the introduction and discussion sections of our manuscript, our goal was to characterise the sequence of folding events that occur during domain synthesis, and the role of Trigger factor.

Reviewer #4:

This is an exciting paper that presents a comprehensive study of DHFR folding on the ribosome. By stopping the protein synthesis at four different locations, the authors are able to sample the foldedness of DHFR as it emerges from the ribosome. An important aspect of this study is the characterization of the role of Trigger factor. The collaborative group of authors encompasses experts in ribosome preparation with stalled nascent chains as well as HDX-MS, the primary approach to characterizing the equilibrium state of the folding intermediates. This work represents a different approach from the usual kinetic trapping that is used for HDX characterization of folding intermediates in which pulse-chase is used to sample the folded states along the kinetic pathway. Here, the authors set-up the folding intermediates by stalling the ribosome and then sample their equilibrium structures. The approach allows them to gain a lot of information about the intermediate folded states of DHFR as it emerges from the ribosome and they do a good job of putting their results in context with other kinetic approaches. They performed all the important control experiments including HDX-MS of the native state DHFR and compare to previous studies on in vitro folding of DHFR from urea denaturation.

I have read the previous reviews and I think the authors did an excellent job of addressing all the reviewer's comments (many of which I agree with). The remaining issues, such as the slight decrease in deuteration over longer time points and the discrepancies in the Max D

deuteration are not uncommon in HDX-MS experiments and do not affect the conclusions of the study.

The results demonstrate several important findings. 1) The folding pathway is different for DHFR emerging from the ribosome as compared to the full-length protein folding from urea. 2) The N-terminal 37 residues of DHFR is more “protected” than would be expected for an unfolded state indicating that it is interacting with the ribosome exit tunnel. 3) By comparing various constructs of the C-terminal domain confined in the exit tunnel or linked to a flexible linker, they show that DHFR instability is induced by unstructured C-termini which is rescued by confinement in the ribosome exit tunnel. 4) DHFR is more active when still attached to the ribosome (a LOT of additional experiments were done to demonstrate statistical significance of this result albeit the difference is only 2-fold).

I think the most exciting part of this work is the characterization of the interactions of Trigger Factor which co-purified with several of the ribosome-nascent chain complexes. The hydrophobic parts of TF apparently weakly dimerize in solution, but when bound to the ribosome they interact with hydrophobic parts of the NC as shown by strong HDX-MS protection in the TF-RNC complex at both hydrophobic and hydrophilic regions of TF. Finally, the authors were able to see protection of the ribosomal proteins lining the exit tunnel demonstrating conclusively that the NC interacts with the exit tunnel surfaces.

We thank the reviewer for their positive assessment of our work, and for taking the time to carefully evaluate the previous reviews/rebuttal.

Final Decision Letter:

Message: 17th Jun 2024

Dear Dr. Balchin,

We are now happy to accept your revised paper "Resolving chaperone-assisted protein folding on the ribosome at the peptide level" for publication as an Article in Nature Structural & Molecular Biology.

Your paper will be published online soon after we receive proof corrections and will appear

in print in the next available issue. You can find out your date of online publication by contacting the production team shortly after sending your proof corrections.

Please note that *Nature Structural & Molecular Biology* is a Transformative Journal (TJ). Authors may publish their research with us through the traditional subscription access route or make their paper immediately open access through payment of an article-processing charge (APC). Authors will not be required to make a final decision about access to their article until it has been accepted. Find out more about Transformative Journals

Authors may need to take specific actions to achieve compliance with funder and institutional open access mandates. If your research is supported by a funder that requires immediate open access (e.g. according to Plan S principles) then you should select the gold OA route, and we will direct you to the compliant route where possible. For authors selecting the subscription publication route, the journal's standard licensing terms will need to be accepted, including self-archiving policies. Those licensing terms will supersede any other terms that the author or any third party may assert apply to any

version of the manuscript.

Sincerely,
Sara

Sara Osman, Ph.D.
Senior Editor
Nature Structural & Molecular Biology